# Evaluating the Fairness of Deep Learning Uncertainty Estimates in Medical Image Analysis

**Raghav Mehta**                                                    RAGHAV@CIM.MCGILL.CA
**Changjian Shui**                                                  MAXSHUI@CIM.MCGILL.CA
**Tal Arbel**                                                       ARBEL@CIM.MCGILL.CA
*Centre for Intelligent Machines, McGill University, Canada*

**Editors:** Accepted for publication at MIDL 2023

## Abstract

Although deep learning (DL) models have shown great success in many medical image analysis tasks, deployment of the resulting models into real clinical contexts requires: (1) that they exhibit robustness and fairness across different sub-populations, and (2) that the confidence in DL model predictions be accurately expressed in the form of uncertainties. Unfortunately, recent studies have indeed shown significant biases in DL models across demographic subgroups (e.g., race, sex, age) in the context of medical image analysis, indicating a lack of fairness in the models. Although several methods have been proposed in the ML literature to mitigate a lack of fairness in DL models, they focus entirely on the absolute performance between groups without considering their effect on uncertainty estimation. In this work, we present the first exploration of the effect of popular fairness models on overcoming biases across subgroups in medical image analysis in terms of bottom-line performance, and their effects on uncertainty quantification. We perform extensive experiments on three different clinically relevant tasks: (i) skin lesion classification, (ii) brain tumour segmentation, and (iii) Alzheimer's disease clinical score regression. Our results indicate that popular ML methods, such as data-balancing and distributionally robust optimization, succeed in mitigating fairness issues in terms of the model performances for some of the tasks. However, this can come at the cost of poor uncertainty estimates associated with the model predictions. This tradeoff must be mitigated if fairness models are to be adopted in medical image analysis.

**Keywords:** Uncertainty, Fairness, Classification, Segmentation, Regression, Brain Tumour, Skin Lesion, Alzheimer's Disease

## 1. Introduction

Deep Learning (DL) models have shown great potential in many clinically relevant applications (e.g. diabetic retinopathy (DR) diagnosis (Gulshan et al., 2016)). Deployment of the resulting models into real-world clinical contexts, and in particular maintaining clinicians' trust, requires that robustness and fairness across different sub-populations are maintained[1]. Unfortunately, several studies have indeed exposed significant biases in DL models across sub-populations (e.g. according to race, sex, age) in the context of medical image analysis (Zong et al., 2022). For example, in Larrazabal et al. (2020), it is shown that a Computer-Assisted Diagnosis system trained on a predominantly male dataset for diagnosing thoracic diseases gives lower performance when tested on female patient images (here,

---

1. Ricci Lara et al. (2022) provides a really good overview of the necessity to address the issue of fairness, potential sources of biases, and the remaining challenges, for machine learning models in medical imaging.

the underrepresented sex). In Burlina et al. (2021), the authors show how data imbalance in the training dataset leads to a disparity in accuracies across sub-populations (dark vs. light skinned individuals) in the diagnosis of DR. Similar issue of racial bias for groups under-represented in the training data is reported for various medical image analysis tasks such as X-ray pathology classification (Seyyed-Kalantari et al., 2021), cardiac MR image segmentation (Puyol-Antón et al., 2021), and brain MR segmentation (Ioannou et al., 2022).

Several methods have been proposed in the machine learning literature to mitigate the lack of fairness (Mehrabi et al., 2021) in the models. This includes data balancing (Japkowicz and Stephen, 2002; Idrissi et al., 2022), which was shown to be successful for some medical imaging contexts (Puyol-Antón et al., 2021; Ioannou et al., 2022). In the machine learning and computer vision fairness literature, the objective is to bridge the performance gap across subgroups with different attributes. It is well established in the literature (Du et al., 2020; Zietlow et al., 2022), however, that fairness across different subgroups can come at the cost of poor overall performance. In those fields, they do not consider the effect of the bias mitigation methods on the uncertainties associated with the model output. In medical image analysis, however, it has been shown that real clinical contexts would benefit from knowledge about the confidence in the model predictions, when made explicit in the form of uncertainties (Band et al., 2021). Specifically, trust would be established should uncertainties associated with the predictions be higher when the model is incorrect, and low where model outputs are correct. Various successful frameworks for quantifying models uncertainties in the context of medical image analysis have been presented for tasks such as image segmentation (Nair et al., 2020; Jungo and Reyes, 2019), image synthesis (Tanno et al., 2021; Mehta and Arbel, 2018), and image classification (Molle et al., 2019; Ghesu et al., 2019). However, these methods only analyze the output uncertainties for the entire population, without consideration of the results for population subgroups.

In this work, we conjecture that uncertainty quantification can help mitigate some potential risks in clinical deployment related to a lack of robustness and fairness for under-represented populations. However, the uncertainties will only help clinicians make more informed decisions if they are accurate. Specifically, a machine learning model that under-performs for an under-represented subgroup should indicate high uncertainties associated with its output for that subgroup. Conversely, a machine learning model that achieves fairness in terms of performance across different subgroups, but produces low uncertainties for predictions where it makes mistakes, would become less trustworthy to clinicians.

In this paper, we present the first analysis of the effect of popular fairness models at overcoming biases of DL models across subgroups for various medical image analysis tasks, and investigate and quantify their effects on the estimated output uncertainties. Specifically, we perform extensive experiments on three different clinically relevant tasks: (i) multi-class skin lesion classification (Codella et al., 2019), (ii) multi-class brain tumour segmentation (Bakas et al., 2018), and (iii) Alzheimer's disease clinical score (Jack Jr et al., 2008) regression. Our results indicate a lack of fairness in model performance for under-represented groups. The uncertainties associated with the outputs behave differently across different groups. We show that popular methods designed to mitigate the lack of fairness, specifically data balancing (Puyol-Antón et al., 2021; Ioannou et al., 2022; Idrissi et al., 2022; Zong et al., 2022) and robust optimization (Sagawa et al., 2019; Zong et al., 2022) do indeed improve fairness for some tasks. However, this comes at the expense of poor

performance of the estimated uncertainties in some cases. This tradeoff must be mitigated if fairness models are to be adopted in medical image analysis.

## 2. Methodology: Fairness in Uncertainty Estimation

This paper aims to evaluate the effectiveness of various popular machine learning fairness models at mitigating biases across subgroups in various medical image analysis contexts in terms of (a) the absolute performance of the models and (b) the uncertainty estimates across the subgroups. Although general, the framework and associated notations focus on binary sensitive attributes (e.g., sex, binarized ages, disease stages).

Consider a dataset $D = \{X, Y, A\} = \{(x_i, y_i, a_i)\}_{i=1}^{N}$ with N total samples. Here, $x_i \in \mathbb{R}^{P \times Q}$ or $x_i \in \mathbb{R}^{P \times Q \times S}$ represents 2D or 3D input image, $y_i$ represents corresponding ground truth labels, and $a_i = \{0, 1\}$ represents the sensitive binary group-attribute. $y_i$ depends on the task at hand: $y_i \in \{0, 1, .., C\}$ for image-level classification, $y_i \in \mathbb{R}$ for image-level regression, and $y_i \in \{0, 1, ..C\}^{P \times Q}$ or $y_i \in \{0, 1, ..C\}^{P \times Q \times S}$ for 2D/3D voxel-level segmentation. The dataset can be further divided into subgroups, $A = \{0, 1\}$, based on the value of the sensitive attribute: (i) $D^0 = \{X^0, Y^0, A = 0\} = \{(x_i^0, y_i^0, a_i = 0)\}_{i=1}^{M}$ and (ii) $D^1 = \{X^1, Y^1, A = 1\} = \{(x_i^1, y_i^1, a_i = 0)\}_{i=1}^{L}$, where $M + L = N$.

Let us consider a deep learning model $f(., \theta)$ that produces a set of outputs $\hat{Y} = f(X, \theta)$ for a set of input images, $X$. The goal here is to define a global fairness metric that is applicable and consistent across a wide variety of tasks (e.g. classification, segmentation, regression). The majority of the fairness metrics (Hinnefeld et al., 2018) are only defined for the classification task. There has been some recent work related to the fairness of segmentation models (Puyol-Antón et al., 2021; Ioannou et al., 2022), where fairness gap metrics are aligned with the one presented in this work. To our knowledge, fairness in medical imaging regression has not yet been explored. Fairness can be defined as follows: A machine learning model is considered to be fair if the difference in the task-specific performance metric between different subgroups is low. To that end, a general fairness gap (FG) metric calculates the differences in the task-specific evaluation metric (EM) values between $\hat{Y}$ and $Y$ conditioned on a binary sensitive attribute $A$.

$$\text{FG}(A = 0, A = 1) = |\text{EM}(Y^0, \hat{Y}^0) - \text{EM}(Y^1, \hat{Y}^1)|. \tag{1}$$

A machine learning model is fair for the sensitive attribute $A$ if $\text{FG}(A = 0, A = 1) = 0$. EM differs depending on the task at hand. Accuracy for image classification, Dice value for segmentation, and mean squared error for image-level regression. EM is calculated for each image separately and then averaged across the dataset for a voxel-level segmentation task. For image classification or regression tasks, EM is calculated directly at a dataset level.

In this work, we focus on Bayesian deep learning (BDL) models (Neal, 2012; Gal and Ghahramani, 2016; Lakshminarayanan et al., 2017; Smith and Gal, 2018), which are widely adopted within the medical image analysis community given their ability to produce uncertainty estimates, $\hat{u}_i$, associated with the model output $\hat{y}_i$. Popular uncertainty estimates include sample variance, predicted variance, entropy, and mutual information (Kendall and Gal, 2017; Gal et al., 2017). Uncertainties $\hat{u}_i$ are typically normalized between 0 (low uncertainty) and 100 (high uncertainty) across the dataset. In the medical image analysis literature, the quality of the estimated uncertainties is evaluated based on the objective

of being correct when confident and highly uncertain when incorrect (Mehta et al., 2022; Nair et al., 2020; Lakshminarayanan et al., 2017). To this end, all predictions whose output uncertainties ($\hat{u}_i$) are above a threshold ($\tau$) are filtered (labeled as uncertain). The EM is calculated on the remaining certain predictions ($\hat{Y}_\tau$ and $Y_\tau$) (below the threshold):

$$\text{FG}_\tau(A = 0, A = 1) = |\text{EM}_\tau(Y_\tau^0, \hat{Y}_\tau^0) - \text{EM}_\tau(Y_\tau^1, \hat{Y}_\tau^1)|. \tag{2}$$

At $\tau = 100$, equations 1 and 2 become equivalent. A higher degree of fairness in uncertainty estimation is established through a reduced fairness gap ($\text{FG}_{\tau 1} \leq \text{FG}_{\tau 2}$) when the number of filtered uncertain predictions increases. In other words, when the uncertainty threshold is reduced ($\tau 1 < \tau 2$), thereby increasing the number of filtered uncertain predictions, the differences in the performances on the remaining confident predictions across the subgroups should be reduced. However, this decrease should not lead to a reduction in overall performance. In other words, it is desirable that $EM_{\tau 1} \geq EM_{\tau 2}$. Conversely, an increase in the fairness gap ($\text{FG}_{\tau 1} > \text{FG}_{\tau 2}$) indicates the undesirable effect of having a higher degree of confidence in incorrect predictions for one of the subgroups.

## 3. Experiments and Results

Extensive experimentation involves comparisons of two established fairness models against a baseline: (i) A **Baseline-Model**: trained on a dataset without consideration of any subgroup information; (ii) A **Balanced-Model**: trained on a dataset where each subgroup contains an equal number of samples during the training, an established baseline fairness model that focuses on mitigating biases due to data imbalance (Puyol-Antón et al., 2021; Ioannou et al., 2022; Idrissi et al., 2022); (iii) A **GroupDRO-Model**: trained with Group-DRO loss (Sagawa et al., 2019) to re-weigh the loss for each subgroup, thereby mitigating lack of fairness through the optimization procedure. The number of images in the test set is the same across all subgroups for fair comparisons.

### 3.1. Multi-class skin lesion classification

Skin cancer is the most prevalent type of cancer in the United States (Guy Jr et al., 2015), which can be diagnosed by classifying skin lesions into different classes.

**Dataset and Sensitive Attribute Rationale:** We use the publicly available International Skin Imaging Collaboration (ISIC) 2019 dataset (Codella et al., 2019) for multi-class skin lesion classification. A dataset of 24947 dermoscopic images is provided, with 8 associated disease scale labels, and with high class imbalance. Demographic patient information (e.g. age, gender) is also provided. We consider age as the sensitive attribute ($a_i$). Following (Zong et al., 2022), the entire dataset is divided into two subsets: patient images with age $\geq 60$ in subgroup $D^0$ with a total of 10805 images, and patients with age $< 60$ in subgroup $D^1$ with a total of 14045 images[2]. The **Baseline-Model** and the **GroupDRO-Model** are trained on a training dataset where subgroup $D^0$ contains 8260 images, while subgroup $D^1$ contains 10892 images. While it appears that subgroup $D^1$ contains approximately 32% more images, it is not strictly the case for all eight classes. A **Balanced-Model** is trained

---

2. We ran experiments with sex as a sensitive attribute, which showed similar results (see Appendix-A.1)

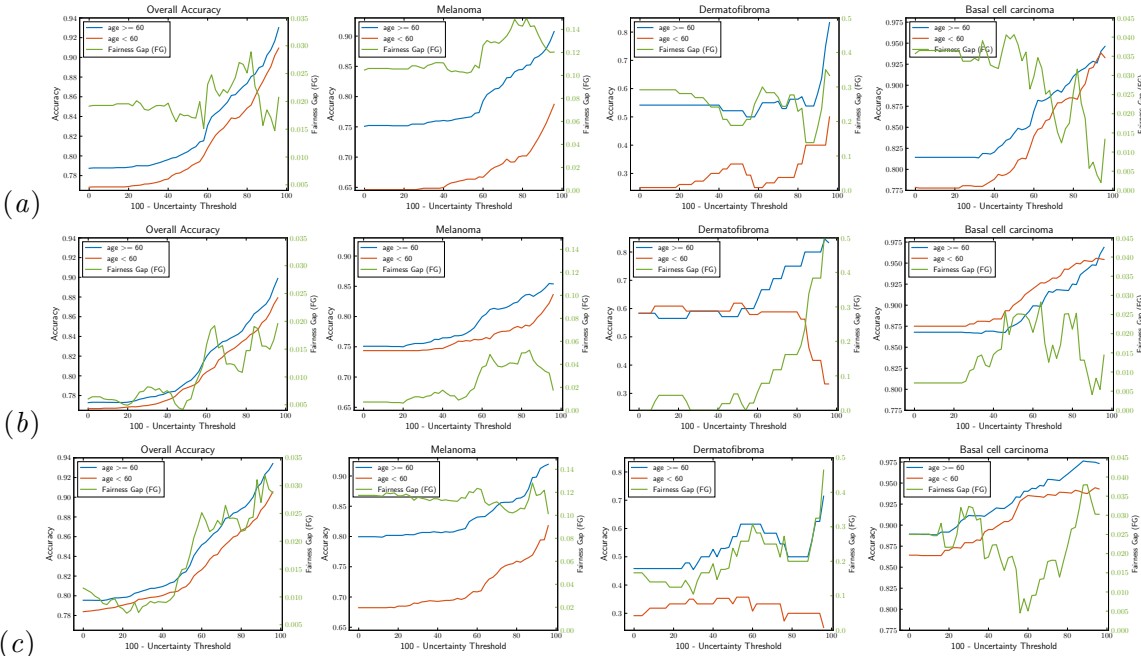

Figure 1: Overall and class-level accuracy (for three classes) against (100 - uncertainty threshold) for (a) **Baseline-Model**, (b) **Balanced-Model**, and (c) **GroupDRO-Model** on the ISIC dataset. Results are shown overall and for each subgroup ($D^0$: age $>= 60$, $D^1$: age $< 60$). For Fairness Gap (FG) refer axis labels on the right.

on a training dataset where both subgroup $D^0$ and subgroup $D^1$ contain 7251 images. Both subgroups are balanced for each of the eight classes of the dataset (but not the same across the eight classes).

**Implementation Details:** An ImageNet pre-trained ResNet-18 (He et al., 2016) model is trained on this dataset. The evaluation metrics (EM) are overall accuracy, overall macro-averaged AUC-ROC, and class-level accuracy. The predictions' uncertainty is measured through the entropy of an Ensemble Dropout model (Smith and Gal, 2018).

**Results:** For the **Baseline-Model**, all four plots in Figure-1(a) show a high fairness gap between the two subgroups when fewer predictions are filtered based on uncertainties (left side of the graph). When filtering more predictions (moving towards the right side of the curve), an increase in the accuracy for each subgroup and a reduction in the fairness gap can be observed. This demonstrates that the model might be incorrect for more images in one of the subgroups, but it usually has *higher uncertainty* in those predictions compared to the other subgroup. Overall Accuracy (Column 1) in Figure-1(b) shows that compared to the **Baseline-Model**, the **Balanced-Model** produces a reduced fairness gap between two subgroups at a low number of filtered predictions (left side of the graph), but at the cost of reduced overall accuracy for each subgroup. The overall accuracy for each subgroup increases with higher uncertainty filtering (towards the right side of the graph). Still, it comes at the expense of a *higher fairness gap.* For classes with a lower number of total images, such as Dermatofibroma in Column 3, filtering out more predictions de-

creases overall performance for one of the subgroups. This shows that while data balancing could enable better fair models at absolute prediction performance level, it comes at the cost of poor uncertainty estimates. Figure-1(c) shows that the **GroupDRO-Model** gives better overall accuracy and better class-wise accuracy compared to the **Baseline-Model** for classes with a high number of total samples (e.g., Melanoma - Column 2, Basal cell carcinoma - Column 4). But it also shows a high fairness gap when a low number of predictions are filtered (left side of the graph). The fairness gap reduces by filtering more predictions. However, it is not completely mitigated for all of the classes. Overall accuracy and classwise-accuracy for classes with a lower number of samples (ex. Dermatofibroma in Column 3) see a marginal increase in the fairness gap with uncertainty-based filtering. Results indicate that the **GroupDRO-Model** might give marginally better absolute performance than the **Baseline-Model**, but it does not produce fair uncertainty estimates across subgroups. Similarly, it can be concluded that different models do not behave consistently across different classes, both in terms of fairness gap and uncertainty evaluation. It indicates that a single model cannot reduce fairness gap and also provide good uncertainty estimation. More results for all eight classes and three models are given in the Appendix-A.

### 3.2. Brain Tumour Segmentation

Automatic segmentation of brain tumours can assist in better and faster diagnosis procedures and surgical planning.

**Dataset and Sensitive Attribute Rationale:** We use the 260 High-Grade Glioma images from the publicly available Brain Tumour Segmentation (BraTS) 2019 challenge dataset (Bakas et al., 2018). The choice for how to split the dataset is based on finding a subgroup where a performance gap is clearly present based on the provided metrics. There can be a number of such subgroups. We initially ran experiments whereby the dataset was split based on imaging centers (i.e. binary subgroups: TCIA vs non-TCIA). Our results, included in the Appendix-B.1, indicated that there is no bias across the resulting groups. It is well established that there is a significant bias in the BraTS dataset, whereby the performance of small tumour segmentation is significantly worse than that of large tumour segmentation. This is an important bias to overcome. The image dataset is therefore divided into two subsets based on the volume of the enhancing tumour: 206 images with volumes $> 7000\text{ml}^3$ in subgroup $D^0$ and 54 images with volumes $\leq 7000\text{ml}^3$ in subgroup $D^1$. **Baseline-Model** and **GroupDRO-Model** are trained on a dataset of 168 samples from $D^0$ and 30 samples from $D^1$. While a **Balanced-Model** is trained on a balanced training set with 30 samples from each subgroup.

**Implementation Details:** A 3D U-Net (Çiçek et al., 2016; Nair et al., 2020) is trained for tumour segmentation. Following the BraTS dataset convention, tumour segmentation performance is evaluated by calculating Dice scores for three different tumour sub-types: enhancing tumor, whole tumor, and tumour core. The predictions' uncertainty is measured through the entropy of an Ensemble Dropout model (Smith and Gal, 2018).

**Results:** Figure 2 shows that both the **Baseline-Model** and the **GroupDRO-Model** perform similarly for whole tumour (WT) across both subgroups, as an increase in Dice and decrease in the fairness gap is observed with filtering of more voxels in the images (go-

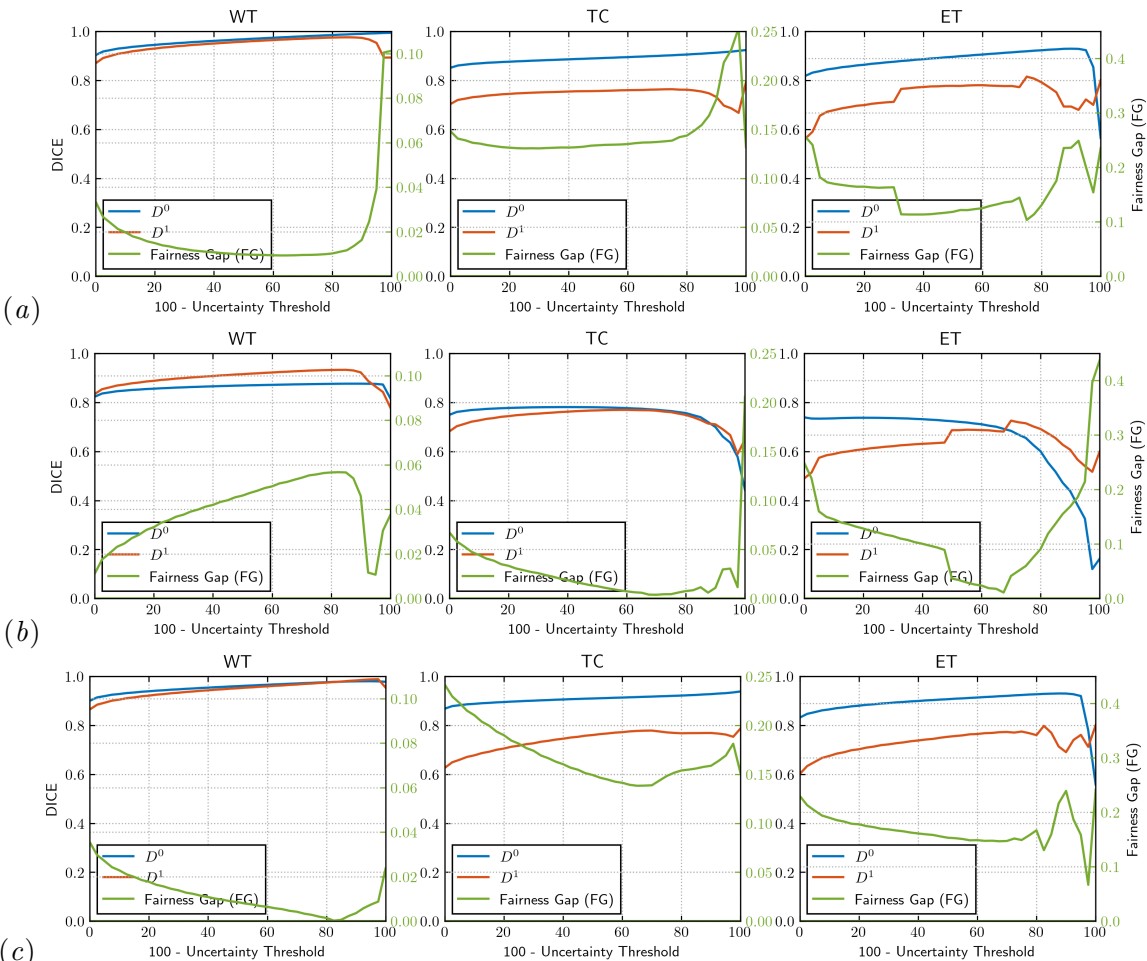

Figure 2: Averaged sample Dice as a function of (100 - uncertainty threshold) for (a) **Baseline-Model**, (b) **Balanced-Model**, and (c) **GroupDRO-Model** on the BraTS dataset. Dice results for whole tumour (WT), tumour core (TC), and enhancing tumour (ET), for both the $D^0$ and $D^1$, set are shown in each column. For Fairness Gap (FG) refer axis labels on the right.

ing from left to right in the graph). For the **Balanced-Model** though initially (left most at an uncertainty threshold of 100) the fairness gap is lower compared to the other two models, it increases with the filtering of more voxels in the images. Tumour core (TC) and enhancing tumour (ET) follow a similar trend, where both the **Baseline-Model** and the **GroupDRO-Model** perform similarly. Although for both TC and ET, the **Balanced-Model** doesn't show an increase in the fairness gap between the two subgroups with a decrease in uncertainty threshold (moving from left to right), a decrease in overall performance for both subgroups is observed. This shows that mitigating the fairness gap by filtering out more voxels is insufficient and may lead to a drop in performance in both subgroups. It can be concluded that for a challenging dataset like BraTS, the **Balanced-Model** or the **GroupDRO-Model** do not produce fair uncertainty estimates across different subgroups.

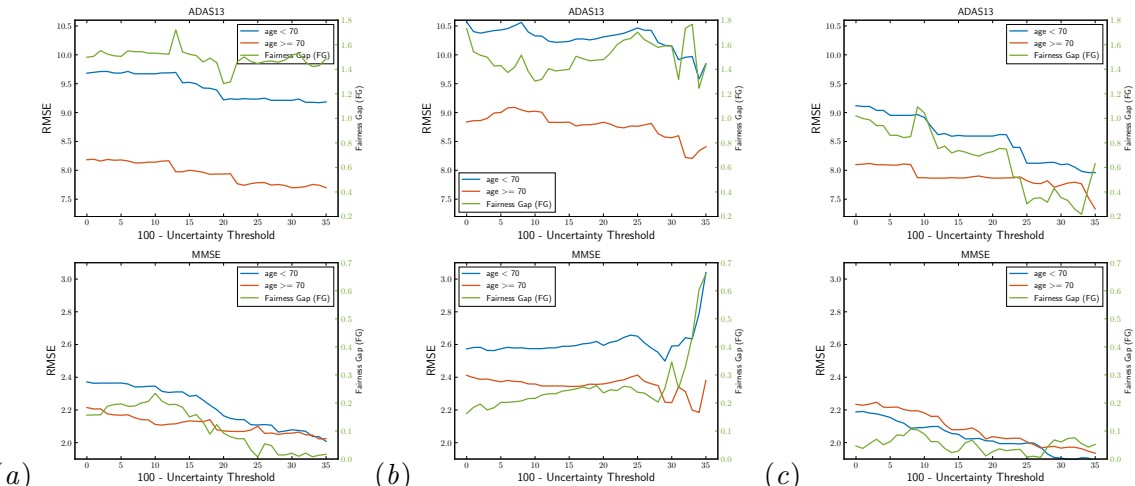

Figure 3: Root Mean Squared Error (RMSE) of ADAS-13 (Top) and MMSE (Bottom) scores as a function of (100-uncertainty threshold) for (a) **Baseline-Model**, (b) **Balanced-Model**, and (c) **GroupDRO-Model** on the ADNI dataset. Specifically, we plot RMSE for each subgroup ($D^0$ with age $< 70$ and $D^1$ with age $\geq 70$). For Fairness Gap (FG) refer axis labels on the right.

### 3.3. Alzheimer's Disease Clinical Score Regression

Alzheimer's disease (AD) is the most common neurodegenerative disorder in elderly people (Goedert and Spillantini, 2006). For AD, clinicians treat symptoms based on structured clinical assessments (e.g., Alzheimer's Disease Assessment Scale – ADAS-13 (Rosen et al., 1984), Mini-Mental State Examination – MMSE (Folstein et al., 1975)).

**Dataset and and Sensitive Attribute Rationale:** Experiments are based on the MRIs of a subset (865 patients) of the Alzheimer's Disease Neuroimaging Initiative (ADNI) dataset (Jack Jr et al., 2008) at different stages of diagnosis: Alzheimer's Disease (145), Mild Cognitive Impairment (498), and Cognitive Normal (222). The dataset also provides demographic patient information such as age and gender. Here, we consider age as a sensitive attribute ($a_i$). The dataset is divided such that patients with age $< 70$ are grouped into $D^0$ (259 patient images), and patients with age $\geq 70$ are grouped into $D^1$ (606 patient images). The threshold for the sensitive attribute was chosen due to the clear performance gap between these subgroups. A **Baseline-Model** and a **GroupDRO-Model** are trained on a dataset that contains 163 samples from $D^0$ and 440 samples from $D^1$. A **Balanced-Model** is trained with 163 samples from each subgroup.

**Implementation Details:** A multi-task 3D ResNet-18 model (Hara et al., 2018) is trained on this dataset to regress ADAS-13 and MMSE scores. Root Mean Squared Error (RMSE) is used as an evaluation metric (EM), where a lower value of RMSE represents better performance. Bayesian Deep Learning model with Ensemble Dropout (Smith and Gal, 2018) is used. A combination of Sample Variance and Predicted Variance, known as total variance (Kendall and Gal, 2017), is used to measure uncertainty associated with the model output.

**Results:**  Figure 3 shows that compared to the **Baseline-Model**, the **Balanced-Model** only marginally decreases the fairness gap in the initial performance between two subgroups, that too at the cost of poor (higher RMSE) absolute performance for each subgroups. The **GroupDRO-Model** shows better absolute performance (lower RMSE) and also a lower fairness gap between each subgroup compared to the other two models. The **Baseline-Model** shows a decrease in the fairness gap between subgroups with a decrease in uncertainty threshold (moving from left to right) for MMSE, but it is not true for ADAS-13. On the contrary, the **Balanced-Model** shows an increase in the fairness gap with a decreased uncertainty threshold for both ADAS-13 and MMSE. The **GroupDRO-Model** gives the best performance as the fairness gap decreases with a decrease in uncertainty threshold.

## 4. Conclusions

In medical image analysis, accurate uncertainty estimates associated with deep learning predictions are necessary for their safe clinical deployment. This paper presented the first exploration of fairness models that mitigate biases across subgroups, and their subsequent effects on uncertainty quantification accuracy. Results on a wide range of experiments for three different tasks indicate that popular fairness methods, such as data balancing and robust optimization, do not work well for all tasks. Furthermore, mitigating fairness in terms of performance can come at the cost of poor uncertainty estimates associated with the outputs. Future work is required to overcome these additional fairness issues prior to clinical deployment of these models. Additional experiments are required to generalize the conclusions presented here, including the exploration of different uncertainty measures (e.g. conformal prediction (Angelopoulos and Bates, 2021)), additional sensitive attributes and associated thresholds, and consideration of multi-class (non binary) attributes.

## Acknowledgments

This investigation was supported by the Natural Sciences and Engineering Research Council (NSERC) of Canada, the Canada Institute for Advanced Research (CIFAR) Artificial Intelligence (AI) Chairs program.

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

# Appendix A. Multi-Class Skin Lesion Classification

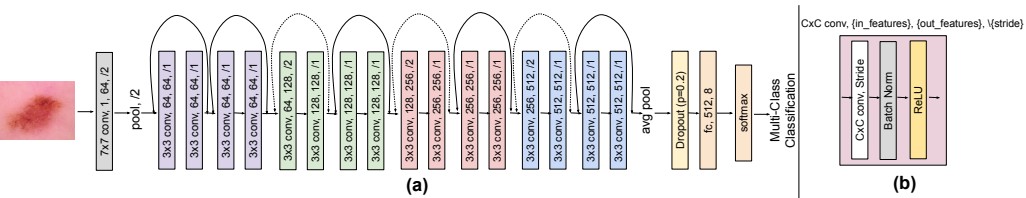

Figure 4: (a) A 2D ResNet-18 architecture consists of a 7x7 convolutional unit, followed by 16 3x3 convolutional units, one dropout layer (p=0.2), and one fully connected layers. The dotted shortcuts increase dimensions. (b) Each convolutional unit consists of one CxC convolutional layer with stride S, followed by Batch Normalization layer (Ioffe and Szegedy, 2015), and a ReLU layer.

**Implementation Details:** An ImageNet pre-trained 2D ResNet18 (He et al., 2016) architecture was used for the ISIC multi-class disease scale classification task. The network architecture is depicted in Figure-4. A Dropout layer (Srivastava et al., 2014) with p=0.2 is introduced before the fully connected (fc) layer. The network was trained to reduce the categorical cross entropy loss. An Adam optimizer (Kingma and Ba, 2014) with a learning rate of 0.0005 and a weight decay of 0.00001 was used to train the network for a total of 100 epochs, and batch size of 64. The learning rate was decayed with a factor of 0.995 after each epoch. All ISIC images were resized to 600x450 size and normalized with mean subtraction and divide by std. Random Horizontal Flip, Random Vertical Flip, and Random rotation in the range of 0-30, was applied as data augmentation on each image. The code was written in PyTorch (Paszke et al., 2019) and ran on Nvidia GeForce RTX 3090 GPU with 24GB memory. For generating EnsembleDropout (Smith and Gal, 2018), we train three different networks with different random initialization of network weights and take 20 MC-Dropout samples (Gal and Ghahramani, 2016) from each. This results in a total of 60 Monte-Carlo samples for each image.

|  | ISIC Dataset | | | | | | | | |
|---|---|---|---|---|---|---|---|---|---|
|  | Melanoma | Melanocytic Nevus | Basal Cell Carcinoma | Actinic Keratosis | Benign Keratosis | Dermatofibroma | Vascular Lesion | Squamous Cell Carcinoma | Total |
| $D^0$ | 2593 | 2600 | 2387 | 707 | 1785 | 79 | 108 | 546 | 10805 |
| $D^1$ | 1844 | 9958 | 930 | 157 | 813 | 160 | 145 | 82 | 14045 |
| **Overall** | 4437 | 12558 | 3317 | 864 | 2598 | 239 | 253 | 628 | 24850 |

Table 1: Number of images for each class and each subgroup for the whole ISIC dataset. From this, we can see a high-class imbalance across different classes. Similarly, distribution across both subgroups for a particular class is also different. For example, while for Melanoma, Basal Cell Carcinoma, Actinic Keratosis, Benign Keratosis, and Squamous Cell Carcinoma, $D^0$ has a higher number of samples compared to $D^1$, for the rest of the classes (Melanocytic Nevus, Dermatofibroma, and Vascular Lesion) $D^1$ has a higher number of samples compared to $D^0$.

| | Training Dataset (Baseline-Model and GroupDRO-Model) | | | | | | | | |
|---|---|---|---|---|---|---|---|---|---|
| | Melanoma | Melanocytic Nevus | Basal Cell Carcinoma | Actinic Keratosis | Benign Keratosis | Dermatofibroma | Vascular Lesion | Squamous Cell Carcinoma | Total |
| $D^0$ | 1835 | 1638 | 1895 | 594 | 1388 | 50 | 68 | 470 | 7938 |
| $D^1$ | 1161 | 8280 | 585 | 99 | 513 | 122 | 100 | 52 | 10912 |
| Overall | 2996 | 9918 | 2480 | 693 | 1901 | 172 | 168 | 522 | 18850 |

Table 2: Number of images for each class and each subgroup for the training dataset used to train the **Baseline-Model** and the **GroupDRO-Model**. Similar to the whole ISIC dataset (Table-1), we see high-class imbalance across different classes, and different distributions across both subgroups for a particular class.

| | Training Dataset (Balanced-Model) | | | | | | | | |
|---|---|---|---|---|---|---|---|---|---|
| | Melanoma | Melanocytic Nevus | Basal Cell Carcinoma | Actinic Keratosis | Benign Keratosis | Dermatofibroma | Vascular Lesion | Squamous Cell Carcinoma | Total |
| $D^0$ | 1161 | 1638 | 585 | 99 | 513 | 50 | 68 | 52 | 4166 |
| $D^1$ | 1161 | 1638 | 585 | 99 | 513 | 50 | 68 | 52 | 4166 |
| Overall | 2322 | 3276 | 1170 | 198 | 1026 | 100 | 136 | 104 | 8332 |

Table 3: Number of images for each class and each subgroup for the training dataset used to train the **Balanced-Model**. Compared to the training dataset used for the **Baseline-Model** and the **GroupDRO-Model** (Table-2), we balance the number of samples across both subgroups, but we do not balance across different classes.

| | Validation Dataset (Baseline-Model, GroupDRO-Model, and Balanced-Model) | | | | | | | | |
|---|---|---|---|---|---|---|---|---|---|
| | Melanoma | Melanocytic Nevus | Basal Cell Carcinoma | Actinic Keratosis | Benign Keratosis | Dermatofibroma | Vascular Lesion | Squamous Cell Carcinoma | Total |
| $D^0$ | 204 | 182 | 212 | 66 | 154 | 5 | 7 | 52 | 882 |
| $D^1$ | 129 | 918 | 65 | 11 | 57 | 14 | 12 | 6 | 1212 |
| Overall | 333 | 1100 | 277 | 77 | 211 | 19 | 19 | 58 | 2094 |

Table 4: Number of images for each class and each subgroup in the Validation dataset for all three models (the **Baseline-Model** and the **GroupDRO-Model**, and the **Balanced-Model**). The distribution of samples across both subgroups and across different classes is similar to the Table-1.

| | Testing Dataset (Baseline-Model, GroupDRO-Model, and Balanced-Model) | | | | | | | | |
|---|---|---|---|---|---|---|---|---|---|
| | Melanoma | Melanocytic Nevus | Basal Cell Carcinoma | Actinic Keratosis | Benign Keratosis | Dermatofibroma | Vascular Lesion | Squamous Cell Carcinoma | Total |
| $D^0$ | 554 | 780 | 280 | 47 | 243 | 24 | 33 | 24 | 1985 |
| $D^1$ | 554 | 780 | 280 | 47 | 243 | 24 | 33 | 24 | 1985 |
| Overall | 1108 | 1560 | 560 | 94 | 486 | 48 | 66 | 48 | 3970 |

Table 5: Number of images for each class and each subgroup in the Testing dataset used to test all three models (the **Baseline-Model** and the **GroupDRO-Model**, and the **Balanced-Model**). The distribution of samples across both subgroups is kept similar, but it is not similar across different classes. We kept similar distribution across both subgroups for a fair comparison of their performance, while the distribution across different classes was not kept similar to reflect real-world scenarios where some classes can be more frequent compared to others.

| Baseline-Model | AUC | Accuracy | Balanced-Accuracy |
|---|---|---|---|
| $D^0$ | 96.46 | 78.74 | 72.64 |
| $D^1$ | 95.71 | 76.83 | 66.76 |
| **Fairness Gap** | 0.75 | 1.91 | 5.88 |

Table 6: Overall metrics (AUC, Accuracy, and Balanced-Accuracy) for a **Baseline-Model** trained on the ISIC dataset at $\tau = 100$.

| Balanced-Model | AUC | Accuracy | Balanced-Accuracy |
|---|---|---|---|
| $D^0$ | 93.91 | 77.28 | 63.70 |
| $D^1$ | 95.09 | 76.68 | 66.87 |
| **Fairness Gap** | 1.18 | 0.60 | 3.17 |

Table 7: Overall metrics (AUC, Accuracy, and Balanced-Accuracy) for a **Balanced-Model** trained on the ISIC dataset at $\tau = 100$.

| GroupDRO-Model | AUC | Accuracy | Balanced-Accuracy |
|---|---|---|---|
| $D^0$ | 96.20 | 79.55 | 71.95 |
| $D^1$ | 95.95 | 78.38 | 71.33 |
| **Fairness Gap** | 0.25 | 1.17 | 0.62 |

Table 8: Overall metrics (AUC, Accuracy, and Balanced-Accuracy) for a **GroupDRO-Model** trained on the ISIC dataset at $\tau = 100$.

| Baseline-Model | Class-level Accuracy | | | | | | | |
|---|---|---|---|---|---|---|---|---|
| | Melanoma | Melanocytic Nevus | Basal Cell Carcinoma | Actinic Keratosis | Benign Keratosis | Dermatofibroma | Vascular Lesion | Squamous Cell Carcinoma |
| $D^0$ | 75.09 | 86.03 | 81.43 | 61.70 | 66.67 | 54.17 | 72.73 | 83.33 |
| $D^1$ | 64.62 | 91.41 | 77.86 | 65.96 | 64.20 | 25.00 | 90.91 | 54.17 |
| **Fairness Gap** | 10.47 | 5.38 | 3.57 | 4.26 | 2.47 | 29.17 | 18.18 | 29.16 |

Table 9: Per class accuracy for a **Baseline-Model** trained on the ISIC dataset at $\tau = 100$.

| Balanced-Model | Class-level Accuracy | | | | | | | |
|---|---|---|---|---|---|---|---|---|
| | Melanoma | Melanocytic Nevus | Basal Cell Carcinoma | Actinic Keratosis | Benign Keratosis | Dermatofibroma | Vascular Lesion | Squamous Cell Carcinoma |
| $D^0$ | 75.09 | 83.33 | 86.78 | 38.30 | 66.26 | 58.33 | 84.85 | 16.67 |
| $D^1$ | 74.37 | 79.87 | 87.50 | 57.45 | 68.72 | 58.33 | 87.88 | 20.83 |
| **Fairness Gap** | 0.72 | 3.46 | 0.72 | 19.15 | 2.46 | 0.00 | 3.03 | 4.16 |

Table 10: Per class accuracy for a **Balanced-Model** trained on the ISIC dataset at $\tau = 100$.

| GroupDRO-Model | Class-level Accuracy | | | | | | | |
|---|---|---|---|---|---|---|---|---|
| | Melanoma | Melanocytic Nevus | Basal Cell Carcinoma | Actinic Keratosis | Benign Keratosis | Dermatofibroma | Vascular Lesion | Squamous Cell Carcinoma |
| $D^0$ | 79.96 | 81.02 | 88.93 | 57.45 | 71.61 | 45.83 | 75.76 | 75.00 |
| $D^1$ | 68.23 | 87.69 | 86.43 | 85.11 | 66.67 | 29.17 | 84.85 | 62.50 |
| **Fairness Gap** | 11.73 | 6.67 | 2.50 | 27.66 | 4.94 | 16.66 | 9.09 | 12.50 |

Table 11: Per class accuracy for a **GroupDRO-Model** trained on the ISIC dataset at $\tau = 100$.

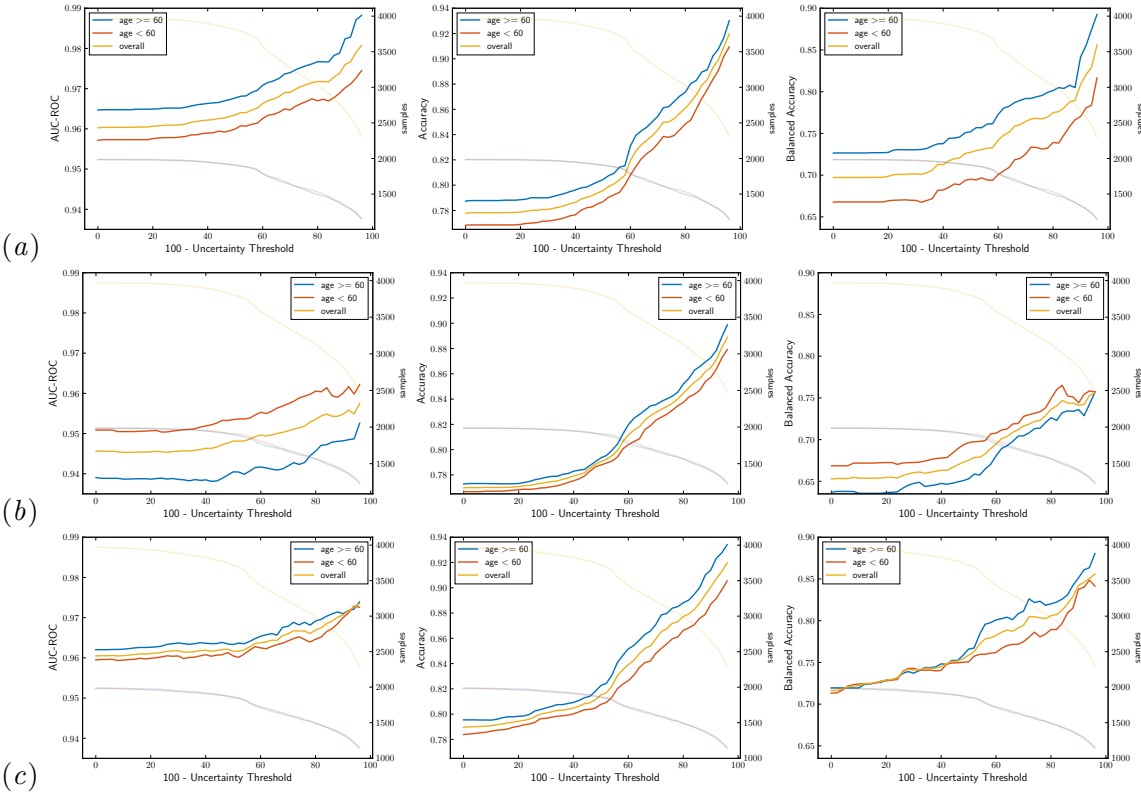

Figure 5: **ISIC:** Overall AUC, accuracy, and Balanced Accuracy as a function of uncertainty threshold for (a) **Baseline-Model**, (b) **Balanced-Model**, and (c) **GroupDRO-Model** on the ISIC dataset. In addition to metrics, the total number of testing images for each subgroup ($D^0$ - age $>= 60$ and $D^1$ - age $< 60$) are shown as light colours.

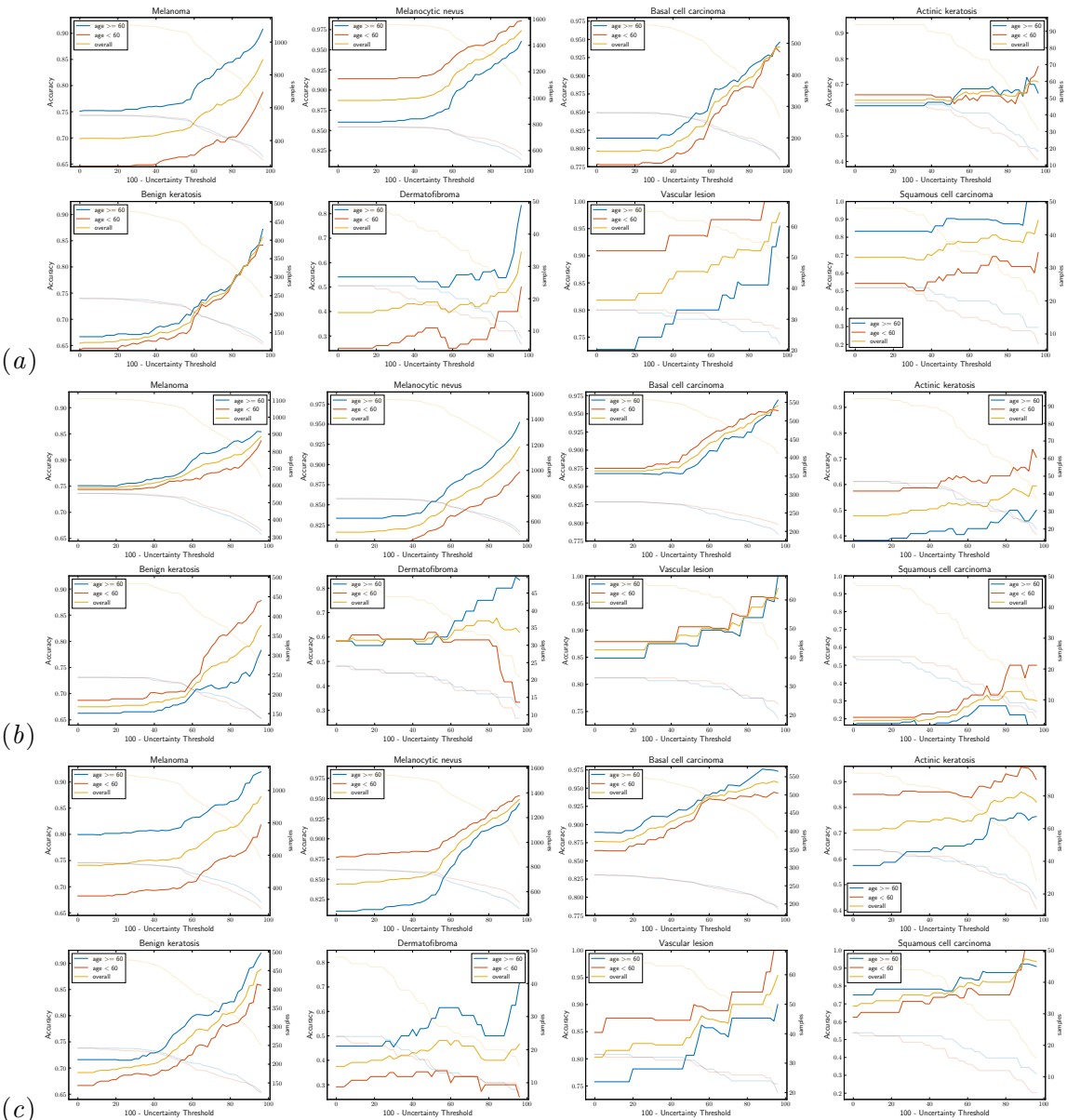

Figure 6: **ISIC:** Class-level accuracy as a function of uncertainty threshold for (a) **Baseline-Model**, (b) **Balanced-Model**, and (c) **GroupDRO-Model** on the ISIC dataset. In addition to the accuracy, the total number of testing images for each subgroup ($D^0$ - age $>= 60$ and $D^1$ - age $< 60$) are shown as light colours.

**A.1. ISIC - Sex as a sensitive attribute**

We use sex as a sensitive attribute for experiments in this section. Specifically, we divide the ISIC dataset into two subsets based on the sex associated with each image (male vs female). The entire dataset is divided into two subsets: patient images from female patients in subgroup $D^0$ with a total of 11661 images, and patient images from male patients in subgroup $D^1$ with a total of 13286 images.

| | ISIC Dataset | | | | | | | | |
|---|---|---|---|---|---|---|---|---|---|
| | Melanoma | Melanocytic Nevus | Basal Cell Carcinoma | Actinic Keratosis | Benign Keratosis | Dermatofibroma | Vascular Lesion | Squamous Cell Carcinoma | Total |
| $D^0$ | 1980 | 6379 | 1317 | 406 | 1134 | 117 | 125 | 203 | 11661 |
| $D^1$ | 2461 | 6225 | 2000 | 458 | 1467 | 122 | 128 | 425 | 13286 |
| Overall | 4441 | 12604 | 3317 | 864 | 2601 | 239 | 253 | 628 | 24947 |

Table 12: Number of images for each class and each subgroup for the whole ISIC dataset. From this, we can see a high-class imbalance across different classes. Similarly, distribution across both subgroups for a particular class is also different. For example, while for Melanoma, Basal Cell Carcinoma, Actinic Keratosis, Benign Keratosis, and Squamous Cell Carcinoma, $D^0$ has a higher number of samples compared to $D^1$, for the rest of the classes (Melanocytic Nevus, Dermatofibroma, and Vascular Lesion) $D^1$ has a higher number of samples compared to $D^0$.

| | Training Dataset (Baseline-Model and GroupDRO-Model) | | | | | | | | |
|---|---|---|---|---|---|---|---|---|---|
| | Melanoma | Melanocytic Nevus | Basal Cell Carcinoma | Actinic Keratosis | Benign Keratosis | Dermatofibroma | Vascular Lesion | Squamous Cell Carcinoma | Total |
| $D^0$ | 1248 | 4061 | 830 | 257 | 715 | 73 | 78 | 128 | 7390 |
| $D^1$ | 1680 | 3922 | 1445 | 303 | 1015 | 78 | 81 | 328 | 8852 |
| Overall | 2928 | 7983 | 2275 | 560 | 1730 | 151 | 159 | 456 | 16242 |

Table 13: Number of images for each class and each subgroup for the training dataset used to train the **Baseline-Model** and the **GroupDRO-Model**. Similar to the whole ISIC dataset (Table-12), we see high-class imbalance across different classes, and different distributions across both subgroups for a particular class.

| | Training Dataset (Balanced-Model) | | | | | | | | |
|---|---|---|---|---|---|---|---|---|---|
| | Melanoma | Melanocytic Nevus | Basal Cell Carcinoma | Actinic Keratosis | Benign Keratosis | Dermatofibroma | Vascular Lesion | Squamous Cell Carcinoma | Total |
| $D^0$ | 1248 | 3922 | 830 | 257 | 715 | 73 | 78 | 128 | 7251 |
| $D^1$ | 1248 | 3922 | 830 | 257 | 715 | 73 | 78 | 128 | 7251 |
| Overall | 2496 | 7844 | 1660 | 514 | 1430 | 146 | 156 | 256 | 14502 |

Table 14: Number of images for each class and each subgroup for the training dataset used to train the **Balanced-Model**. Compared to the training dataset used for the **Baseline-Model** and the **GroupDRO-Model** (Table-13), we balance the number of samples across both subgroups, but we do not balance across different classes.

| | Validation Dataset (Baseline-Model, GroupDRO-Model, and Balanced-Model) | | | | | | | | |
|---|---|---|---|---|---|---|---|---|---|
| | Melanoma | Melanocytic Nevus | Basal Cell Carcinoma | Actinic Keratosis | Benign Keratosis | Dermatofibroma | Vascular Lesion | Squamous Cell Carcinoma | Total |
| $D^0$ | 138 | 451 | 92 | 28 | 79 | 8 | 9 | 14 | 819 |
| $D^1$ | 187 | 436 | 160 | 34 | 112 | 8 | 9 | 36 | 982 |
| Overall | 325 | 887 | 252 | 62 | 191 | 16 | 18 | 50 | 1801 |

Table 15: Number of images for each class and each subgroup in the Validation dataset for all three models (the **Baseline-Model** and the **GroupDRO-Model**, and the **Balanced-Model**). The distribution of samples across both subgroups and across different classes is similar to the Table-12.

| | Testing Dataset (Baseline-Model, GroupDRO-Model, and Balanced-Model) | | | | | | | | |
|---|---|---|---|---|---|---|---|---|---|
| | Melanoma | Melanocytic Nevus | Basal Cell Carcinoma | Actinic Keratosis | Benign Keratosis | Dermatofibroma | Vascular Lesion | Squamous Cell Carcinoma | Total |
| $D^0$ | 594 | 1867 | 395 | 121 | 340 | 36 | 38 | 61 | 3452 |
| $D^1$ | 594 | 1867 | 395 | 121 | 340 | 36 | 38 | 61 | 3452 |
| Overall | 1188 | 3734 | 790 | 242 | 680 | 72 | 76 | 122 | 6904 |

Table 16: Number of images for each class and each subgroup in the Testing dataset used to test all three models (the **Baseline-Model** and the **GroupDRO-Model**, and the **Balanced-Model**). The distribution of samples across both subgroups is kept similar, but it is not similar across different classes. We kept similar distribution across both subgroups for a fair comparison of their performance, while the distribution across different classes was not kept similar to reflect real-world scenarios where some classes can be more frequent compared to others.

| Baseline-Model | AUC | Accuracy | Balanced-Accuracy |
|---|---|---|---|
| $D^0$ | 96.24 | 83.02 | 71.77 |
| $D^1$ | 96.83 | 83.02 | 70.23 |
| **Fairness Gap** | 0.59 | 0.00 | 1.54 |

Table 17: Overall metrics (AUC, Accuracy, and Balanced-Accuracy) for a **Baseline-Model** trained on the ISIC dataset at $\tau = 100$.

| Balanced-Model | AUC | Accuracy | Balanced-Accuracy |
|---|---|---|---|
| $D^0$ | 96.26 | 82.24 | 70.26 |
| $D^1$ | 95.92 | 81.66 | 69.42 |
| **Fairness Gap** | 0.34 | 0.58 | 0.74 |

Table 18: Overall metrics (AUC, Accuracy, and Balanced-Accuracy) for a **Balanced-Model** trained on the ISIC dataset at $\tau = 100$.

| GroupDRO-Model | AUC | Accuracy | Balanced-Accuracy |
|---|---|---|---|
| $D^0$ | 95.76 | 80.56 | 70.25 |
| $D^1$ | 96.31 | 80.59 | 69.90 |
| **Fairness Gap** | 0.55 | 0.03 | 0.35 |

Table 19: Overall metrics (AUC, Accuracy, and Balanced-Accuracy) for a **GroupDRO-Model** trained on the ISIC dataset at $\tau = 100$.

| Baseline-Model | Class-level Accuracy | | | | | | | |
|---|---|---|---|---|---|---|---|---|
| | Melanoma | Melanocytic Nevus | Basal Cell Carcinoma | Actinic Keratosis | Benign Keratosis | Dermatofibroma | Vascular Lesion | Squamous Cell Carcinoma |
| $D^0$ | 65.32 | 91.64 | 85.06 | 61.16 | 80.29 | 63.88 | 71.05 | 55.74 |
| $D^1$ | 73.91 | 91.27 | 87.09 | 52.07 | 68.53 | 44.44 | 92.11 | 52.46 |
| Fairness Gap | 8.59 | 0.37 | 2.03 | 9.09 | 11.76 | 19.44 | 21.06 | 3.28 |

Table 20: Per class accuracy for a **Baseline-Model** trained on the ISIC dataset at $\tau = 100$.

| Balanced-Model | Class-level Accuracy | | | | | | | |
|---|---|---|---|---|---|---|---|---|
| | Melanoma | Melanocytic Nevus | Basal Cell Carcinoma | Actinic Keratosis | Benign Keratosis | Dermatofibroma | Vascular Lesion | Squamous Cell Carcinoma |
| $D^0$ | 62.79 | 92.34 | 88.35 | 59.50 | 70.88 | 66.67 | 78.95 | 42.62 |
| $D^1$ | 68.52 | 90.95 | 87.59 | 54.55 | 65.88 | 61.11 | 94.74 | 32.79 |
| Fairness Gap | 5.73 | 1.39 | 0.76 | 4.95 | 5.00 | 5.56 | 15.79 | 9.83 |

Table 21: Per class accuracy for a **Balanced-Model** trained on the ISIC dataset at $\tau = 100$.

| GroupDRO-Model | Class-level Accuracy | | | | | | | |
|---|---|---|---|---|---|---|---|---|
| | Melanoma | Melanocytic Nevus | Basal Cell Carcinoma | Actinic Keratosis | Benign Keratosis | Dermatofibroma | Vascular Lesion | Squamous Cell Carcinoma |
| $D^0$ | 55.05 | 92.07 | 83.29 | 66.12 | 70.29 | 55.56 | 78.95 | 60.66 |
| $D^1$ | 63.13 | 90.52 | 82.28 | 59.50 | 68.24 | 50.00 | 81.58 | 63.93 |
| Fairness Gap | 8.08 | 1.55 | 1.01 | 6.62 | 2.05 | 5.56 | 2.63 | 3.27 |

Table 22: Per class accuracy for a **GroupDRO-Model** trained on the ISIC dataset at $\tau = 100$.

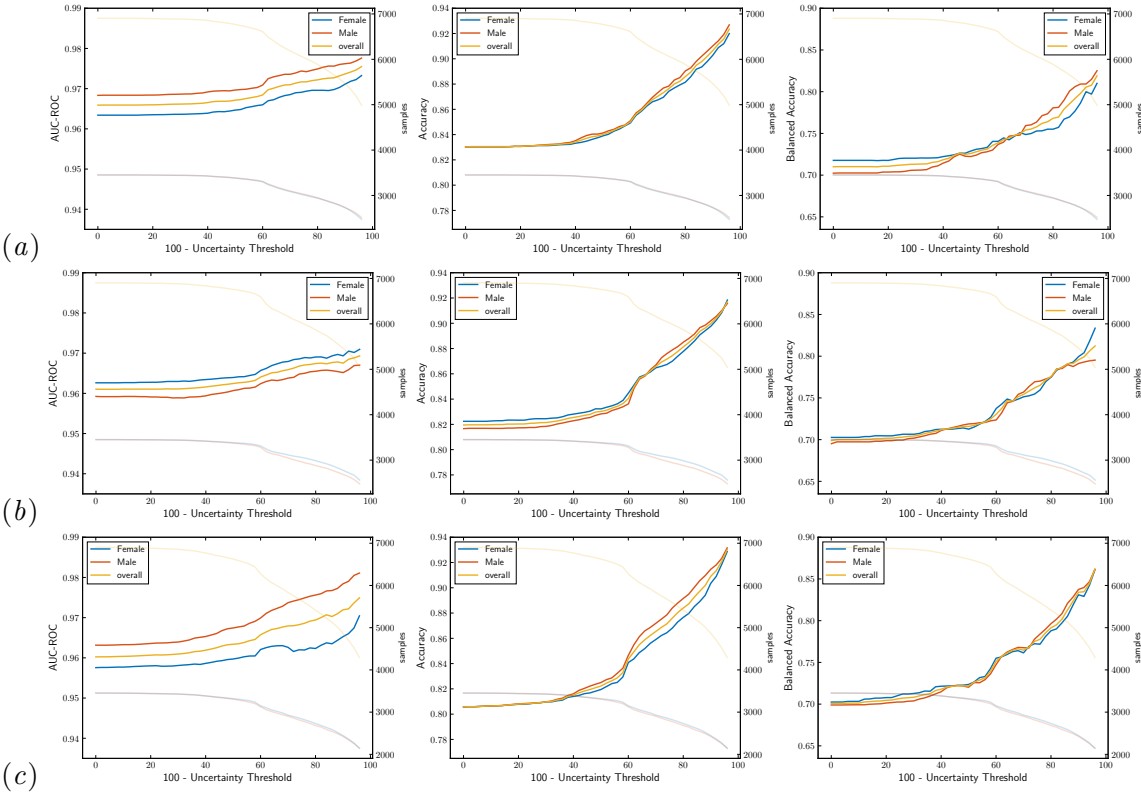

Figure 7: **ISIC-Sex:** Overall AUC, accuracy, and Balanced Accuracy as a function of uncertainty threshold for (a) **Baseline-Model**, (b) **Balanced-Model**, and (c) **GroupDRO-Model** on the ISIC dataset. In addition to metrics, the total number of testing images for each subgroup ($D^0$ - Female and $D^1$ - Male) are shown as light colours.

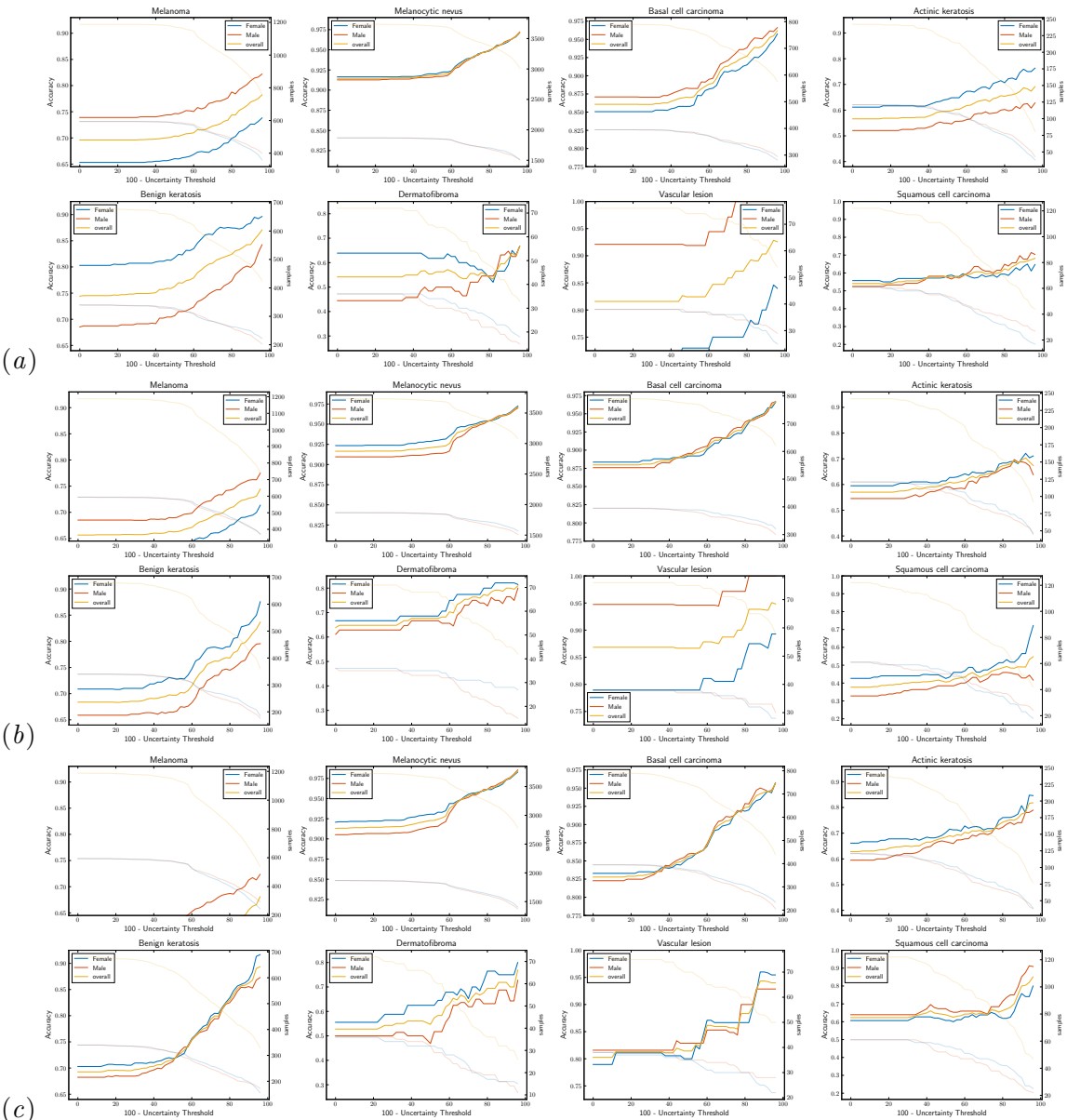

Figure 8: **ISIC-Sex:** Class-level accuracy as a function of uncertainty threshold for (a) **Baseline-Model**, (b) **Balanced-Model**, and (c) **GroupDRO-Model** on the ISIC dataset. In addition to the accuracy, the total number of testing images for each subgroup ($D^0$ - Female and $D^1$ - Male) are shown as light colours.

## Appendix B. Brain Tumour Segmentation

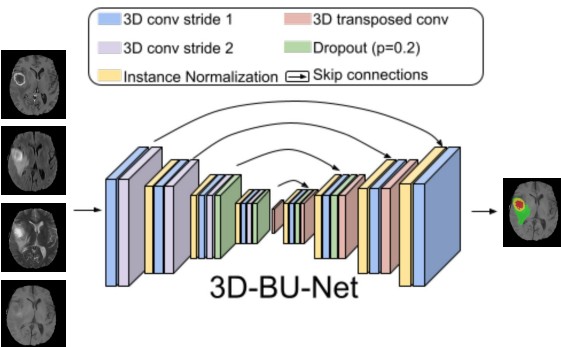

Figure 9: Network architecture diagram of the modified 3D-BU-Net (Nair et al., 2020), used for the multi-class brain tumour segmentation. It takes multi-modal MR images as input and produces multi brain tumour segmentation on the BraTS dataset.

**Implementation Details:** We use a BU-Net (Nair et al., 2020) architecture for brain tumour segmentation on the BraTS dataset. Similar to the original 3D BU-Net, the network consists of encoder and decoder paths that embed convolution and deconvolution operations. High-resolution features from the encoder path were combined with the up-sampled output of the decoder to preserve high-resolution features. Each convolution was followed by rectified linear unit activation (ReLU). Instead of using the batch-normalization layer used in the original U-Net, we used instance normalization (Ulyanov et al., 2016). Instance normalization typically improves performance for small batch sizes. The network was trained using Adam (Kingma and Ba, 2014) optimizer with a learning rate of 0.0002 and weight decay of 0.00001 for a total of 240 epochs to minimize weighted cross-entropy loss. Here, the weights are defined such that the weight increases whenever there are fewer voxels in a particular class. After every epoch, class weights were decayed with a factor of 0.95, which results in equally weighted binary cross-entropy after around 50 epochs. The code was written in PyTorch (Paszke et al., 2019) and ran on Nvidia GeForce RTX 3090 GPU with 24GB memory. For generating EnsembleDropout (Smith and Gal, 2018), we train three different networks with different random initialization of network weights and take 20 MC-Dropout samples (Gal and Ghahramani, 2016) from each. This results in a total of 60 Monte-Carlo samples for each image.

| | Training Set | | Validation Set | Testing Set | BraTS Dataset |
|---|---|---|---|---|---|
| | **Baseline-Model and GroupDRO-Model** | **Balanced Model** | | | |
| $D^0$ | 168 | 30 | 18 | 20 | 206 |
| $D^1$ | 30 | 30 | 4 | 20 | 54 |
| **Overall** | 198 | 60 | 22 | 40 | 260 |

Table 23: Number of samples in both $D^0$ and $D^1$ subgroups for five different datasets: (i) Training Dataset used to train the **Baseline-Model** and the **GroupDRO-Model**, (ii) Training Dataset used to the train the **Balanced-Model**, (iii) Validation set for all three models, (iv) Testing set for all three models, and (v) for the whole BraTS dataset. We can observe that for the BraTS dataset, there is a high disparity between the number of samples for both subgroups.

| Baseline-Model | Dice | | | QU-BraTS Metric | | |
|---|---|---|---|---|---|---|
| | **Whole Tumour** | **Tumour Core** | **Enhancing Tumour** | **Whole Tumour** | **Tumour Core** | **Enhancing Tumour** |
| $D^0$ | 90.34 | 85.14 | 81.80 | 92.50 | 88.81 | 82.90 |
| $D^1$ | 86.99 | 70.33 | 56.13 | 89.37 | 76.11 | 75.05 |
| **Fairness Gap** | 3.35 | 14.81 | 25.67 | 3.13 | 12.70 | 7.85 |

Table 24: Dice (at $\tau = 100$) and QU-BraTS metric (Mehta et al., 2022) values for Whole Tumour, Tumour Core, and Enhancing Tumour of a **Baseline-Model** on the BraTS dataset.

| Balanced-Model | Dice | | | QU-BraTS Metric | | |
|---|---|---|---|---|---|---|
| | **Whole Tumour** | **Tumour Core** | **Enhancing Tumour** | **Whole Tumour** | **Tumour Core** | **Enhancing Tumour** |
| $D^0$ | 82.33 | 74.94 | 73.92 | 86.67 | 77.68 | 71.08 |
| $D^1$ | 83.45 | 68.19 | 48.96 | 86.24 | 74.44 | 71.93 |
| **Fairness Gap** | 1.12 | 6.75 | 24.96 | 0.43 | 3.24 | 0.85 |

Table 25: Dice (at $\tau = 100$) and QU-BraTS metric (Mehta et al., 2022) values for Whole Tumour, Tumour Core, and Enhancing Tumour of a **Balanced-Model** on the BraTS dataset.

| GroupDRO-Model | Dice | | | QU-BraTS Metric | | |
|---|---|---|---|---|---|---|
| | **Whole Tumour** | **Tumour Core** | **Enhancing Tumour** | **Whole Tumour** | **Tumour Core** | **Enhancing Tumour** |
| $D^0$ | 90.02 | 86.80 | 83.16 | 92.07 | 89.84 | 89.33 |
| $D^1$ | 86.47 | 62.64 | 60.14 | 90.30 | 77.42 | 73.17 |
| **Fairness Gap** | 3.55 | 24.16 | 23.02 | 1.77 | 12.42 | 16.16 |

Table 26: Dice (at $\tau = 100$) and QU-BraTS metric (Mehta et al., 2022) values for Whole Tumour, Tumour Core, and Enhancing Tumour of a **GroupDRO-Model** on the BraTS dataset.

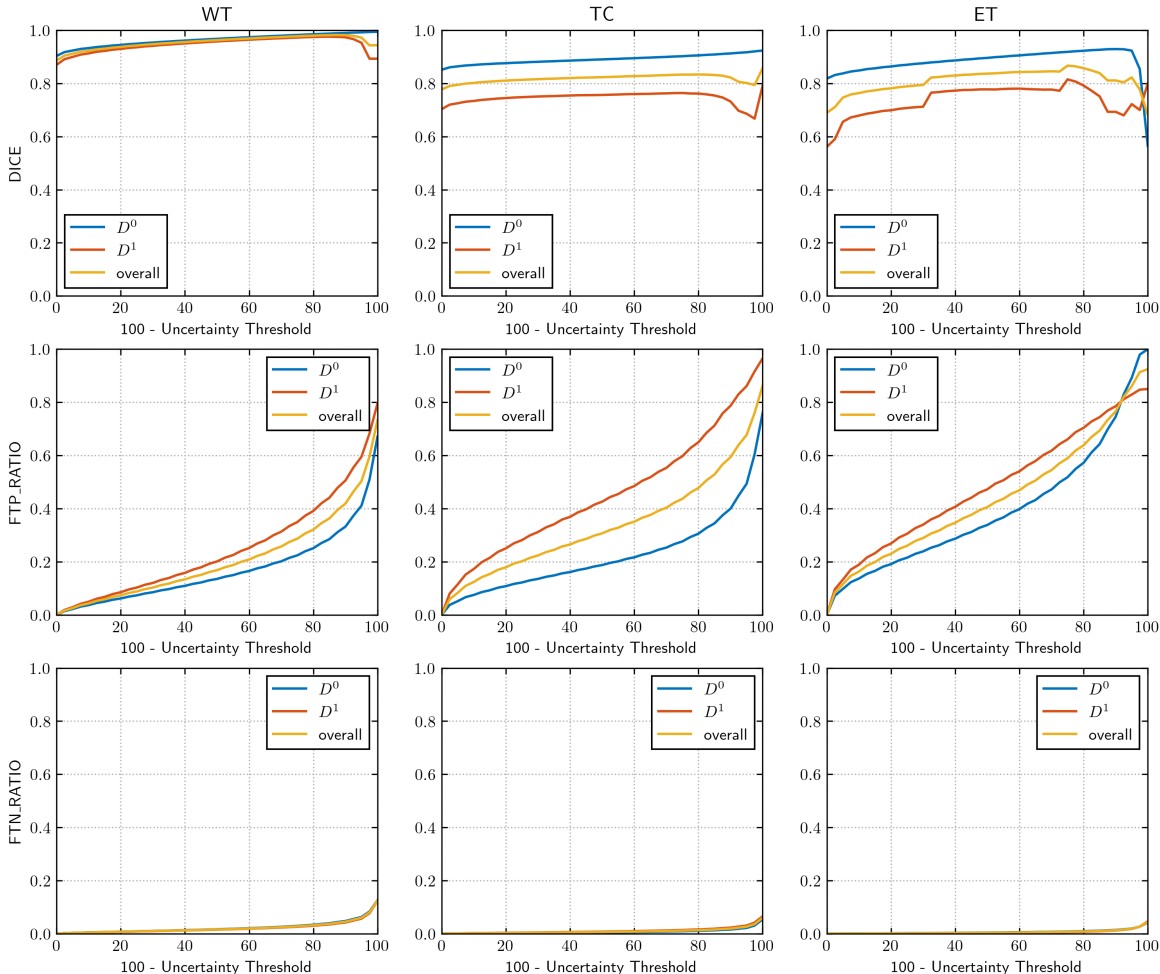

Figure 10: **BraTS:** Dice, Filtered True Positive Ratio (FTP), and Filtered True Negative Ratio (FTN) as a function of uncertainty threshold for **Baseline-Model** on the BraTS dataset. Specifically, we plot Whole Tumour (WT), Tumour Core (TC), and Enhancing Tumour (ET) QU-BraTS (Mehta et al., 2022) metrics for both the $D^0$ and $D^1$ set.

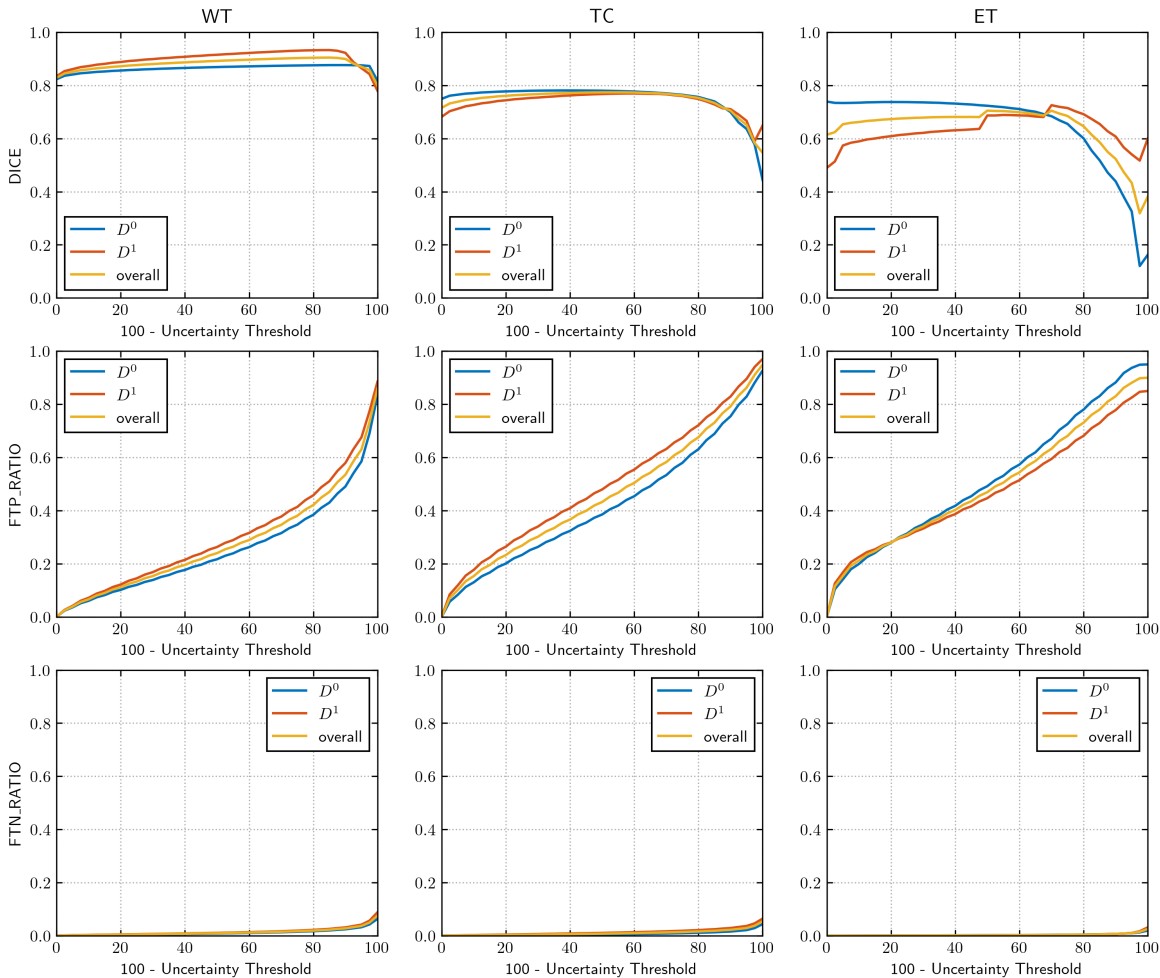

Figure 11: **BraTS:** Dice, Filtered True Positive Ratio (FTP), and Filtered True Negative Ratio (FTN) as a function of uncertainty threshold for **Balanced-Model** on the BraTS dataset. Specifically, we plot Whole Tumour (WT), Tumour Core (TC), and Enhancing Tumour (ET) QU-BraTS (Mehta et al., 2022) metrics for both the $D^0$ and $D^1$ set..

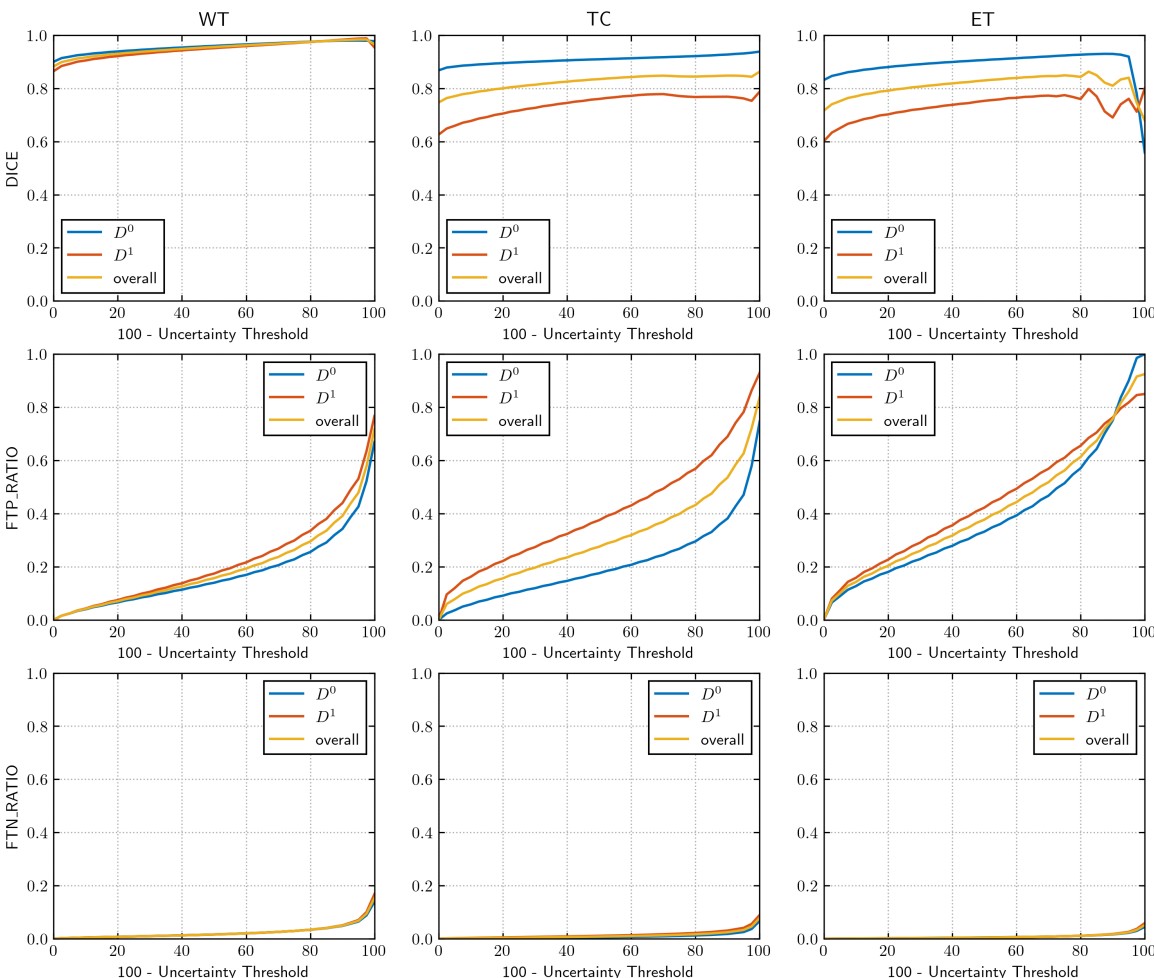

Figure 12: **BraTS:** We plot Dice, Filtered True Positive Ratio (FTP), and Filtered True Negative Ratio (FTN) as a function of uncertainty threshold for **GroupDRO-Model** on the BraTS dataset. Specifically, we plot Whole Tumour (WT), Tumour Core (TC), and Enhancing Tumour (ET) QU-BraTS (Mehta et al., 2022) metrics for both the $D^0$ and $D^1$ set.

### B.1. Brain Tumour Segmentation - Experiment and Results for imaging centers based fairness and uncertainty evaluation

In this section, We use the 260 High-Grade Glioma (HGG) images from the publicly available Brain Tumour Segmentation (BraTS) 2019 challenge dataset. The image dataset is divided into two subsets based on the imaging center. Specifically, images coming from TCIA subset were considered in subgroup $D^0$, while images from the rest of the imaging center were considered in subgroup $D^1$. A Baseline-Model and a GroupDRO-Model are trained on a dataset of 74 samples from $D^0$ and 124 samples from $D^1$. While a Balanced-Model is trained on a balanced training set with 74 samples from each subgroup.

| | Training Set | | Validation Set | Testing Set | BraTS Dataset |
|---|---|---|---|---|---|
| | Baseline-Model and GroupDRO-Model | Balanced Model | | | |
| $D^0$ | 74 | 74 | 8 | 20 | 102 |
| $D^1$ | 124 | 74 | 14 | 20 | 158 |
| **Overall** | 198 | 148 | 22 | 40 | 260 |

Table 27: Number of samples in both $D^0$ and $D^1$ subgroups for five different datasets: (i) Training Dataset used to train the **Baseline-Model** and the **GroupDRO-Model**, (ii) Training Dataset used to the train the **Balanced-Model**, (iii) Validation set for all three models, (iv) Testing set for all three models, and (v) for the whole BraTS dataset. We can observe that for the BraTS dataset, there is a high disparity between the number of samples for both subgroups.

| Baseline-Model | Dice | | | QU-BraTS Metric | | |
|---|---|---|---|---|---|---|
| | Whole Tumour | Tumour Core | Enhancing Tumour | Whole Tumour | Tumour Core | Enhancing Tumour |
| $D^0$ | 91.11 | 88.42 | 84.26 | 93.38 | 91.79 | 84.85 |
| $D^1$ | 91.34 | 86.35 | 83.84 | 92.92 | 90.18 | 85.16 |
| **Fairness Gap** | 0.23 | 2.07 | 0.42 | 0.46 | 1.61 | 0.31 |

Table 28: Dice (at $\tau = 100$) and QU-BraTS metric (Mehta et al., 2022) values for Whole Tumour, Tumour Core, and Enhancing Tumour of a **Baseline-Model** on the BraTS dataset.

| Balanced-Model | Dice | | | QU-BraTS Metric | | |
|---|---|---|---|---|---|---|
| | Whole Tumour | Tumour Core | Enhancing Tumour | Whole Tumour | Tumour Core | Enhancing Tumour |
| $D^0$ | 90.49 | 88.28 | 83.73 | 92.96 | 91.18 | 86.16 |
| $D^1$ | 91.23 | 83.78 | 81.79 | 92.95 | 89.08 | 85.64 |
| **Fairness Gap** | 0.74 | 4.50 | 1.94 | 0.01 | 2.10 | 0.52 |

Table 29: Dice (at $\tau = 100$) and QU-BraTS metric (Mehta et al., 2022) values for Whole Tumour, Tumour Core, and Enhancing Tumour of a **Balanced-Model** on the BraTS dataset.

| GroupDRO-Model | Dice | | | QU-BraTS Metric | | |
|---|---|---|---|---|---|---|
| | Whole Tumour | Tumour Core | Enhancing Tumour | Whole Tumour | Tumour Core | Enhancing Tumour |
| $D^0$ | 90.45 | 87.63 | 83.84 | 92.35 | 91.03 | 84.38 |
| $D^1$ | 91.79 | 85.35 | 83.39 | 93.13 | 90.21 | 85.97 |
| **Fairness Gap** | 1.34 | 2.28 | 0.45 | 0.78 | 0.72 | 1.59 |

Table 30: Dice (at $\tau = 100$) and QU-BraTS metric (Mehta et al., 2022) values for Whole Tumour, Tumour Core, and Enhancing Tumour of a **GroupDRO-Model** on the BraTS dataset.

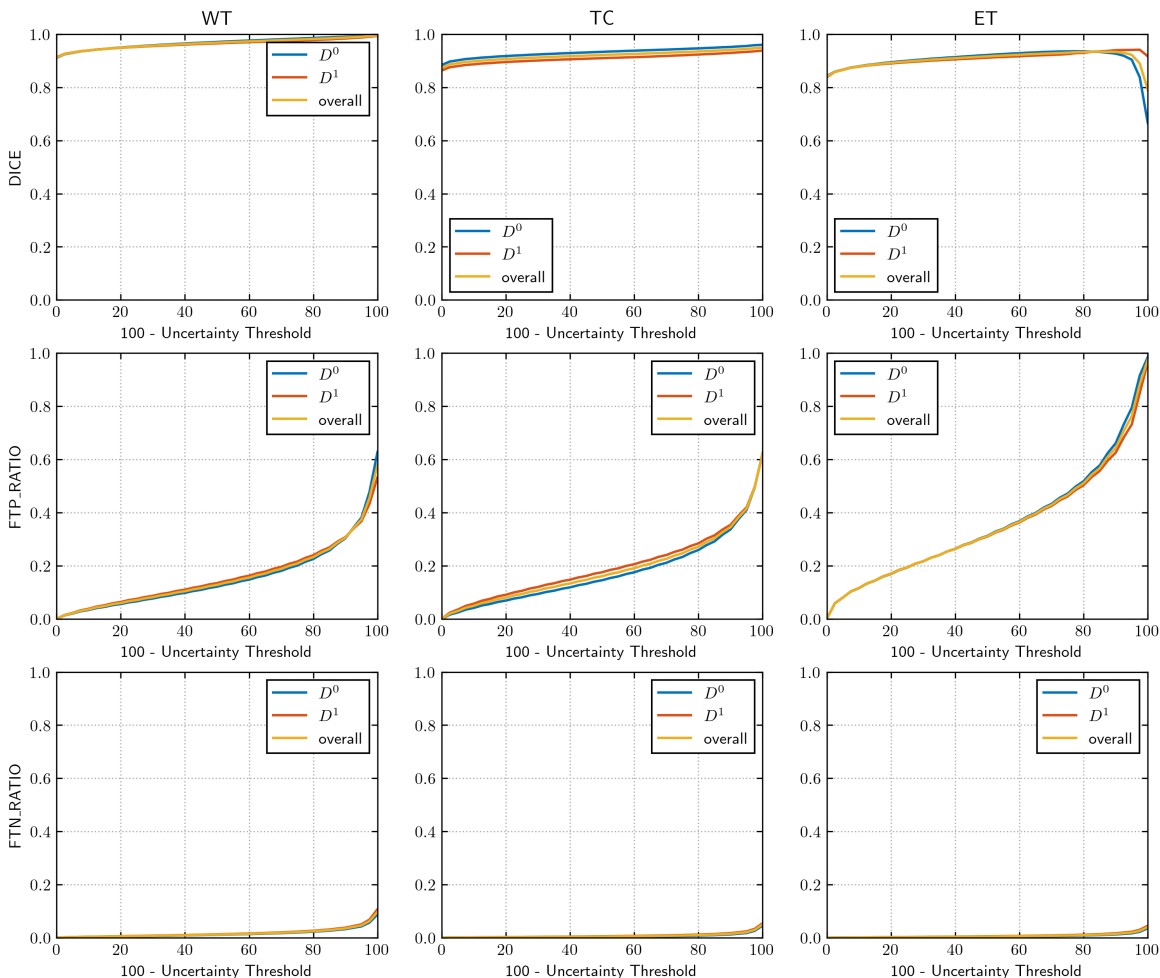

Figure 13: **BraTS-Imaging-Centre:** Dice, Filtered True Positive Ratio (FTP), and Filtered True Negative Ratio (FTN) as a function of uncertainty threshold for **Baseline-Model** on the BraTS dataset. Specifically, we plot Whole Tumour (WT), Tumour Core (TC), and Enhancing Tumour (ET) QU-BraTS (Mehta et al., 2022) metrics for both the $D^0$ and $D^1$ set.

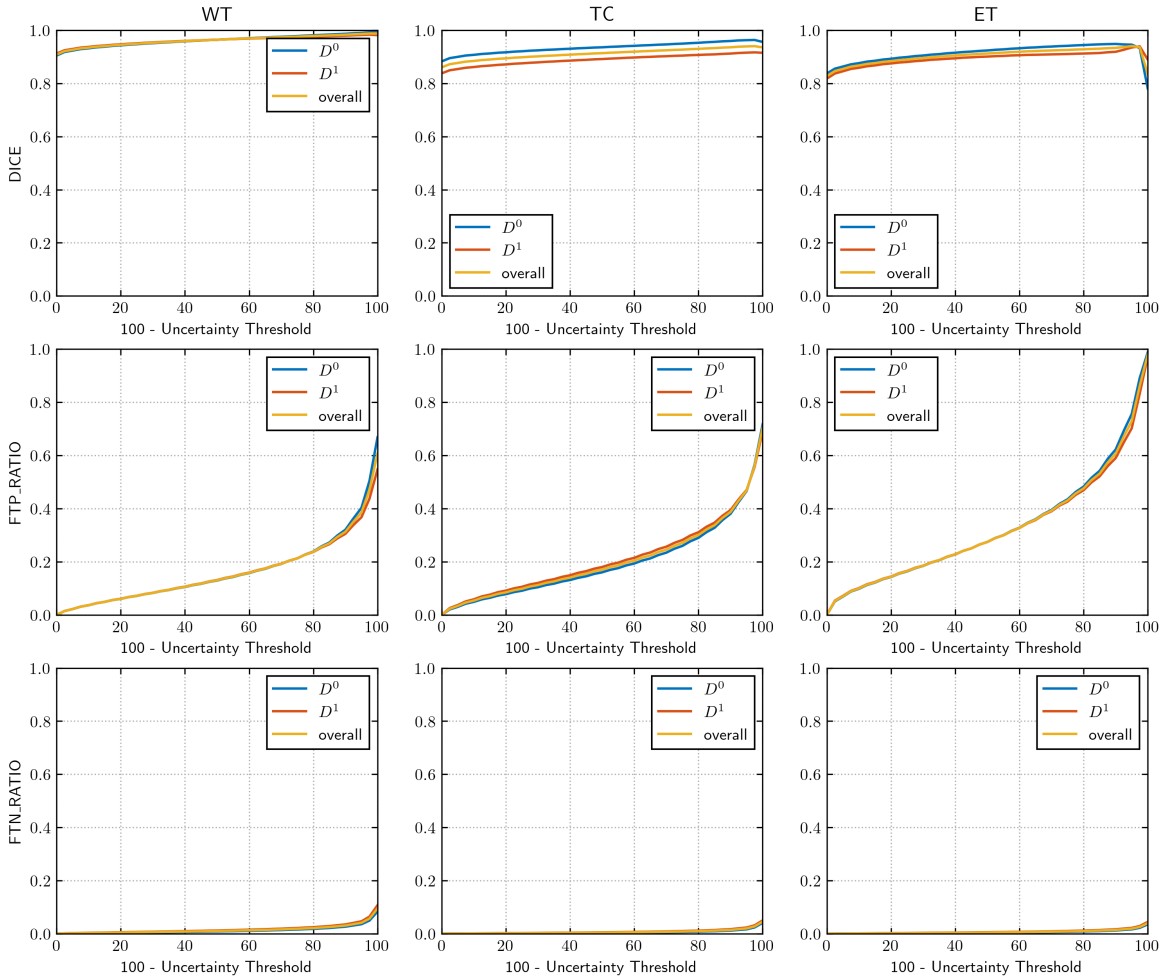

Figure 14: **BraTS-Imaging-Centre:** Dice, Filtered True Positive Ratio (FTP), and Filtered True Negative Ratio (FTN) as a function of uncertainty threshold for **Balanced-Model** on the BraTS dataset. Specifically, we plot Whole Tumour (WT), Tumour Core (TC), and Enhancing Tumour (ET) QU-BraTS (Mehta et al., 2022) metrics for both the $D^0$ and $D^1$ set.

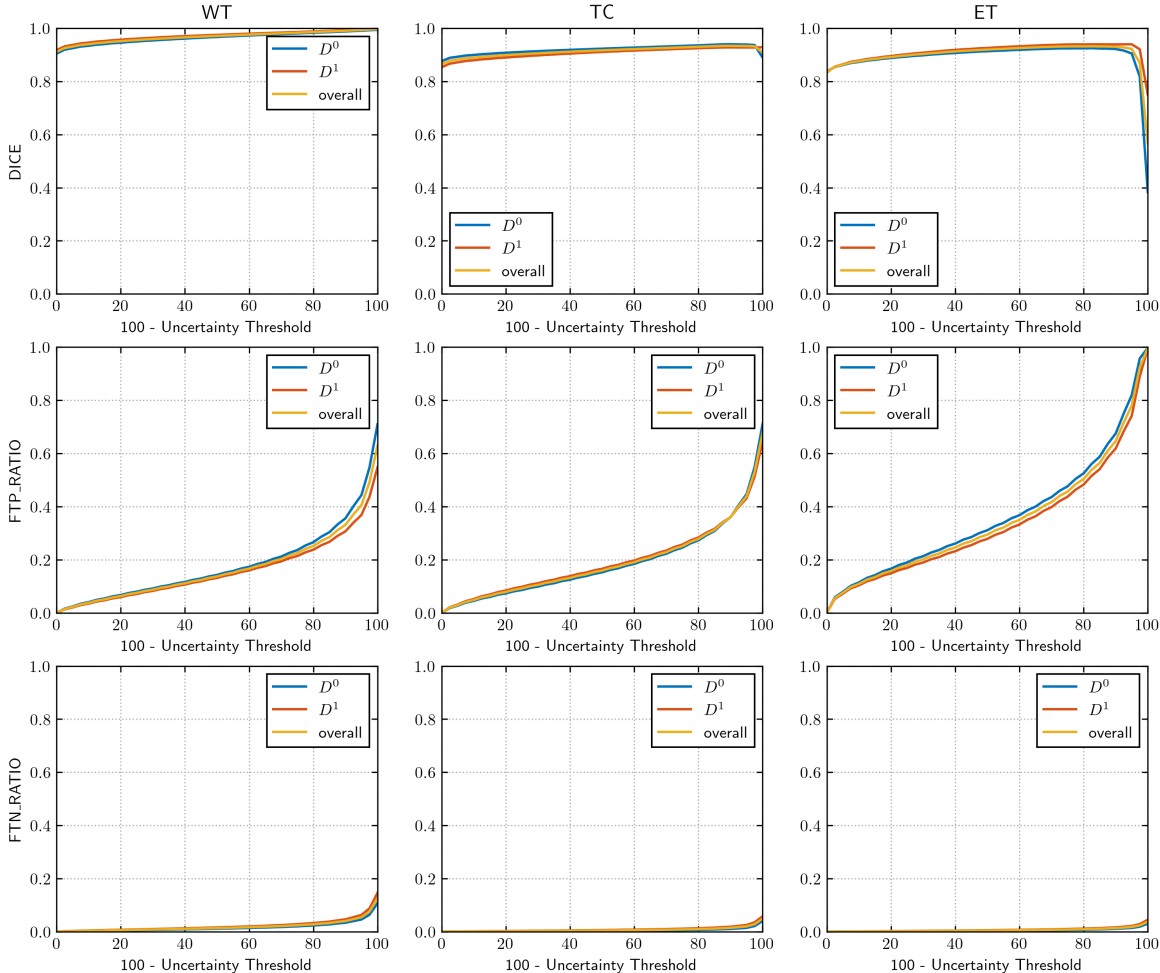

Figure 15: **BraTS-Imaging-Centre:** We plot Dice, Filtered True Positive Ratio (FTP), and Filtered True Negative Ratio (FTN) as a function of uncertainty threshold for **GroupDRO-Model** on the BraTS dataset. Specifically, we plot Whole Tumour (WT), Tumour Core (TC), and Enhancing Tumour (ET) QU-BraTS (Mehta et al., 2022) metrics for both the $D^0$ and $D^1$ set.

## Appendix C. Alzheimer's Disease Clinical Score Regression

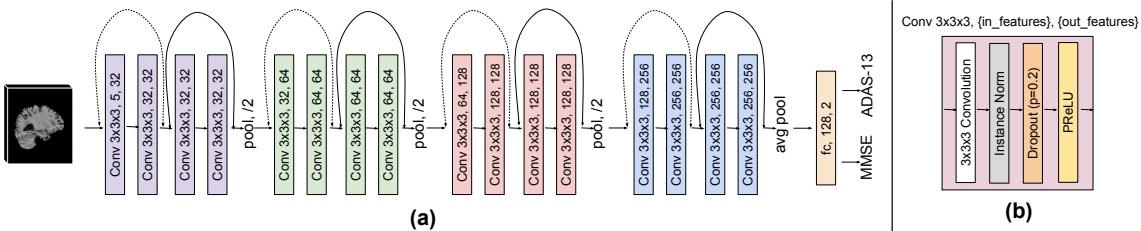

Figure 16: Network architecture diagram of modified 3D-ResNet-18 (Hara et al., 2018) for the Alzheimer's Disease clinical regression pipeline for predicting both ADAS-13 and MMSE scores. The network takes 3D T1-weighted MR image as input.

**Implementation Details:** A 3D ResNet34 (Hara et al., 2018) architecture was designed for the task of clinical score prediction [3]. The network was modified to be a multi-task network, such that it predicts both ADAS-13 and MMSE scores simultaneously. The network was trained to reduce the combined mean squared error losses for both ADAS-13 and MMSE. An Adam optimizer with a learning rate of 0.0002 and a weight decay of 0.00001 was used to train the network for a total of 200 epochs. The learning rate was decayed with a factor of 0.995 after each epoch. The code was written in PyTorch (Paszke et al., 2019) and ran on Nvidia GeForce RTX 3090 GPU with 24GB memory. For generating EnsembleDropout (Smith and Gal, 2018), we train three different networks with different random initialization of network weights and take 20 MC-Dropout samples (Gal and Ghahramani, 2016) from each. This results in a total of 60 Monte-Carlo samples for each image. As the total number of images is low in this dataset, we run the same experiments on five different folds and aggregate their results.

| | ADNI Dataset | | | |
| --- | --- | --- | --- | --- |
| | **AD** | **MCI** | **CN** | **Total** |
| $D^0$ | 33 | 187 | 39 | 259 |
| $D^1$ | 112 | 311 | 183 | 606 |
| **Overall** | 145 | 498 | 222 | 865 |

Table 31: Number of images for each disease stage (AD, MCI, and CN) and each subgroup for the whole ADNI dataset. From this, we can see a high disparity between the total number of samples in each disease stage. Similarly, distribution across subgroups for a particular disease stage is also different.

---

3. https://github.com/kenshohara/3D-ResNets-PyTorch/blob/master/models/resnet.py

| | Training Dataset (Baseline-Model and GroupDRO-Model) | | | |
|---|---|---|---|---|
| | AD | MCI | CN | Total |
| $D^0$ | 15 | 130 | 18 | 163 |
| $D^1$ | 80 | 220 | 140 | 440 |
| Overall | 95 | 350 | 158 | 603 |

Table 32: Number of images for each disease stage (AD, MCI, and CN) and each subgroup for the training dataset used to train the **Baseline-Model** and the **GroupDRO-Model**. Similar to the whole ADNI dataset (Table-31), a high disparity between the total number of samples in each disease stage. Similarly, distribution across subgroups for a particular disease stage is also different.

| | Training Dataset (Balanced-Model) | | | |
|---|---|---|---|---|
| | AD | MCI | CN | Total |
| $D^0$ | 15 | 130 | 18 | 163 |
| $D^1$ | 15 | 130 | 18 | 163 |
| Overall | 30 | 260 | 36 | 326 |

Table 33: Number of images for each disease stage (AD, MCI, and CN) and each subgroup for the training dataset used to train the **Balanced-Model**. Compared to the training dataset used for the **Baseline-Model** and the **GroupDRO-Model** (Table-32), we balance the number of samples across both subgroups for each disease stage, but not across disease stages.

| | Validation Dataset | | | |
|---|---|---|---|---|
| | AD | MCI | CN | Total |
| $D^0$ | 5 | 7 | 19 | 31 |
| $D^1$ | 19 | 29 | 53 | 101 |
| Overall | 24 | 36 | 72 | 132 |

Table 34: Number of images for each disease stage (AD, MCI, and CN) and each subgroup in the Validation dataset for all three models (**Baseline-Model**, **GroupDRO-Model**, and **Balanced-Model**). The distribution of samples across both subgroups and across different disease stages is similar to the Table-31.

| | Testing Dataset | | | |
|---|---|---|---|---|
| | AD | MCI | CN | Total |
| $D^0$ | 13 | 14 | 38 | 65 |
| $D^1$ | 13 | 14 | 38 | 65 |
| Overall | 26 | 28 | 76 | 130 |

Table 35: Number of images for each disease stage (AD, MCI, and CN) and each subgroup in the Testing dataset used to test all three models ( **Baseline-Model**, **GroupDRO-Model**, and **Balanced-Model**). The distribution of samples across both subgroups is kept similar, but it is not similar across different disease stages. We kept similar distribution across both subgroups for a fair comparison of their performance, while the distribution across different disease stages was not kept similar to reflect real-world scenarios where some disease stage can occur more frequently compared to others.

| Baseline-Model | ADAS-13 | | | | | | | |
| --- | --- | --- | --- | --- | --- | --- | --- | --- |
| | RMSE | | | | MAE | | | |
| | All | AD | MCI | CN | All | AD | MCI | CN |
| $D^0$ | 9.68 | 16.25 | 6.88 | 8.93 | 7.60 | 13.61 | 5.66 | 7.92 |
| $D^1$ | 8.18 | 12.82 | 6.01 | 7.98 | 6.33 | 10.84 | 4.71 | 6.67 |
| Fairness Gap | 1.50 | 3.43 | 0.87 | 0.94 | 1.27 | 2.77 | 0.95 | 1.25 |

Table 36: Root Mean Squared Error - RMSE and Mean Absolute Error - MAE (at $\tau = 100$) for ADAS-13 score for All , Alzheimer's (AD), Mild-Cognitive Impairment (MCI), and Cognitive Normal (CN) samples of a **Baseline-Model** trained on the ADNI dataset.

| Balanced-Model | ADAS-13 | | | | | | | |
| --- | --- | --- | --- | --- | --- | --- | --- | --- |
| | RMSE | | | | MAE | | | |
| | All | AD | MCI | CN | All | AD | MCI | CN |
| $D^0$ | 10.57 | 17.25 | 7.13 | 10.20 | 8.54 | 14.86 | 6.01 | 9.54 |
| $D^1$ | 8.84 | 13.49 | 6.86 | 8.33 | 6.94 | 11.44 | 5.31 | 7.34 |
| Fairness Gap | 1.73 | 3.76 | 0.26 | 1.87 | 1.59 | 3.42 | 0.70 | 2.19 |

Table 37: Root Mean Squared Error - RMSE and Mean Absolute Error - MAE (at $\tau = 100$) for ADAS-13 score for All , Alzheimer's (AD), Mild-Cognitive Impairment (MCI), and Cognitive Normal (CN) samples of a **Balanced-Model** trained on the ADNI dataset.

| Balanced-Model | ADAS-13 | | | | | | | |
| --- | --- | --- | --- | --- | --- | --- | --- | --- |
| | RMSE | | | | MAE | | | |
| | All | AD | MCI | CN | All | AD | MCI | CN |
| $D^0$ | 9.12 | 15.47 | 6.63 | 7.73 | 7.11 | 12.96 | 5.39 | 7.73 |
| $D^1$ | 8.10 | 12.08 | 6.25 | 8.10 | 6.26 | 9.56 | 5.07 | 8.10 |
| Fairness Gap | 1.02 | 3.39 | 0.38 | 0.37 | 0.85 | 3.40 | 0.32 | 0.37 |

Table 38: Root Mean Squared Error - RMSE and Mean Absolute Error - MAE (at $\tau = 100$) for ADAS-13 score for All , Alzheimer's (AD), Mild-Cognitive Impairment (MCI), and Cognitive Normal (CN) samples of a **GroupDRO-Model** trained on the ADNI dataset.

| Baseline-Model | MMSE | | | | | | | |
|---|---|---|---|---|---|---|---|---|
| | **RMSE** | | | | **MAE** | | | |
| | **All** | **AD** | **MCI** | **CN** | **All** | **AD** | **MCI** | **CN** |
| $D^0$ | 2.37 | 4.43 | 1.57 | 1.69 | 1.79 | 3.89 | 1.26 | 1.52 |
| $D^1$ | 2.21 | 3.45 | 1.72 | 2.00 | 1.75 | 2.86 | 1.40 | 1.72 |
| **Fairness Gap** | 0.16 | 0.99 | 0.15 | 0.31 | 0.03 | 1.03 | 0.14 | 0.20 |

Table 39: Root Mean Squared Error - RMSE and Mean Absolute Error - MAE (at $\tau = 100$) for MMSE score for All , Alzheimer's (AD), Mild-Cognitive Impairment (MCI), and Cognitive Normal (CN) samples of a **Baseline-Model** trained on the ADNI dataset.

| Balanced-Model | MMSE | | | | | | | |
|---|---|---|---|---|---|---|---|---|
| | **RMSE** | | | | **MAE** | | | |
| | **All** | **AD** | **MCI** | **CN** | **All** | **AD** | **MCI** | **CN** |
| $D^0$ | 2.57 | 4.10 | 1.87 | 2.44 | 2.11 | 3.56 | 1.58 | 2.44 |
| $D^1$ | 2.41 | 3.14 | 1.96 | 2.73 | 2.00 | 2.52 | 1.64 | 2.48 |
| **Fairness Gap** | 0.16 | 0.96 | 0.09 | 0.29 | 0.12 | 1.04 | 0.06 | 0.04 |

Table 40: Root Mean Squared Error - RMSE and Mean Absolute Error - MAE (at $\tau = 100$) for MMSE score for All , Alzheimer's (AD), Mild-Cognitive Impairment (MCI), and Cognitive Normal (CN) samples of a **Balanced-Model** trained on the ADNI dataset.

| GroupDRO-Model | MMSE | | | | | | | |
|---|---|---|---|---|---|---|---|---|
| | **RMSE** | | | | **MAE** | | | |
| | **All** | **AD** | **MCI** | **CN** | **All** | **AD** | **MCI** | **CN** |
| $D^0$ | 2.19 | 3.88 | 1.68 | 1.52 | 1.63 | 3.07 | 1.33 | 1.34 |
| $D^1$ | 2.23 | 3.09 | 1.91 | 2.15 | 1.73 | 2.43 | 1.51 | 1.73 |
| **Fairness Gap** | 0.05 | 0.79 | 0.23 | 0.63 | 0.10 | 0.64 | 0.18 | 0.39 |

Table 41: Root Mean Squared Error - RMSE and Mean Absolute Error - MAE (at $\tau = 100$) for MMSE score for All , Alzheimer's (AD), Mild-Cognitive Impairment (MCI), and Cognitive Normal (CN) samples of a **GroupDRO-Model** trained on the ADNI dataset.

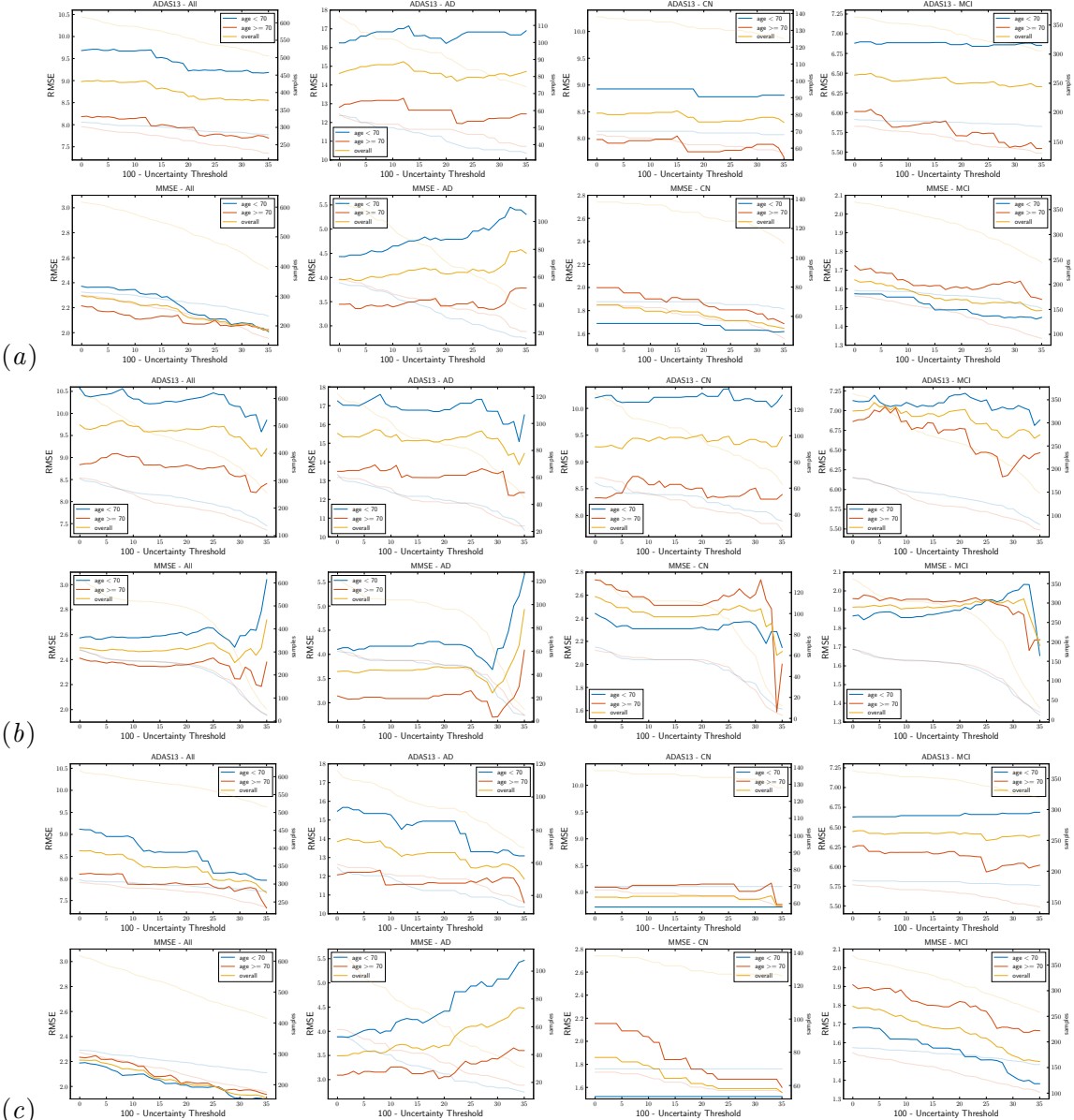

Figure 17: **ADNI:** Root Mean Squared Error (RMSE) of ADAS-13 (Top) and MMSE (Bottom) score prediction tasks as a function of uncertainty threshold for (a) **Baseline-Model**, (b) **Balanced-Model**, and (c) **GroupDRO-Model** on the ADNI dataset. Specifically, we plot RMSE for all samples as well as samples for each of the disease stages (AD, MCI, and CN) in each subgroup ($D^0$ - age $< 70$ and $D^1$ - age $\geq 70$). The total number of samples as a function of uncertainty thresholds in are depicted with light colours in these plots.

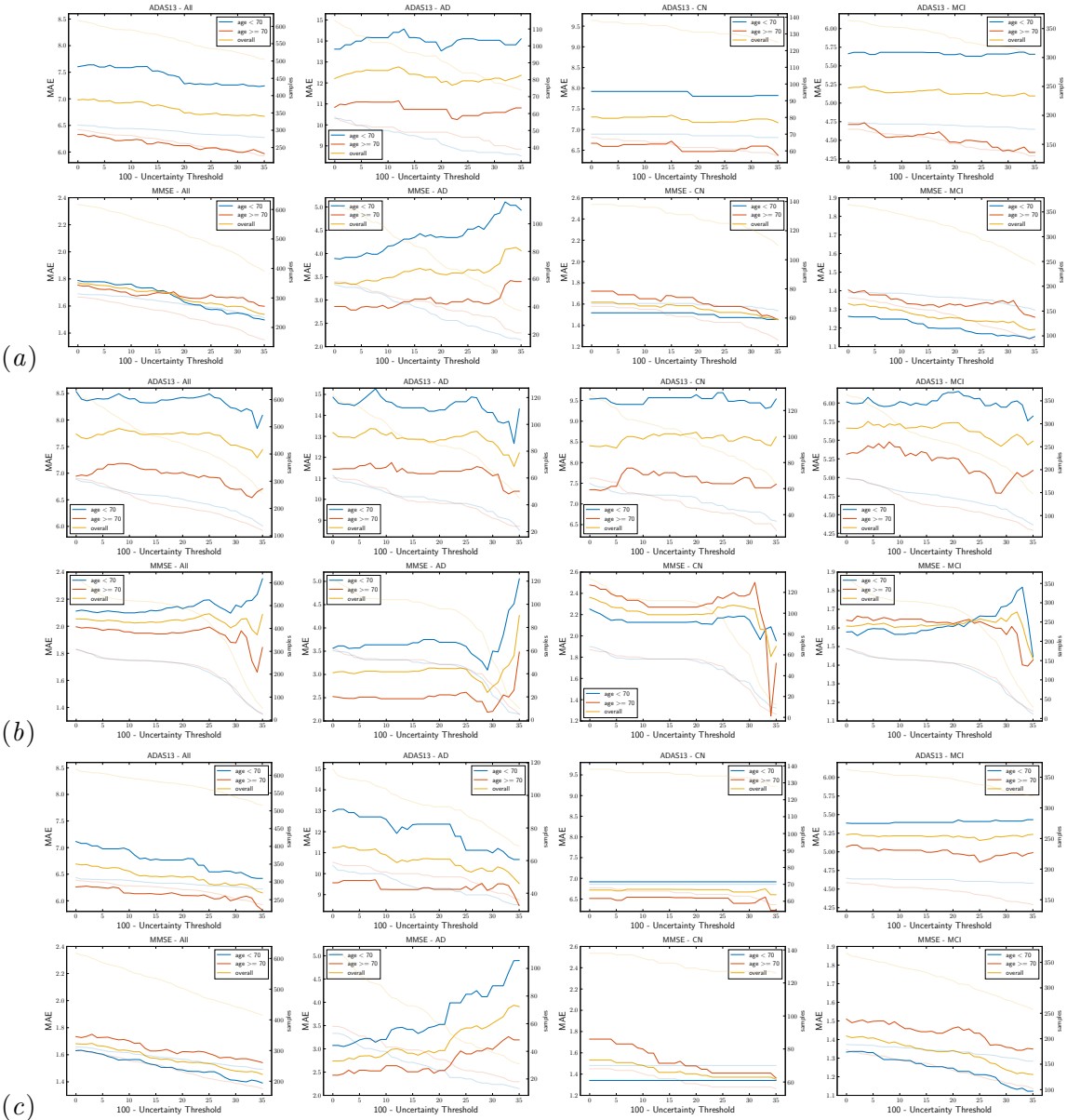

Figure 18: **ADNI:** Mean Absolute Error (MAE) of ADAS-13 (Top) and MMSE (Bottom) score prediction tasks as a function of uncertainty threshold for (a) **Baseline-Model**, (b) **Balanced-Model**, and (c) **GroupDRO-Model** on the ADNI dataset. Specifically, we plot RMSE for all samples as well as samples for each of the disease stages (AD, MCI, and CN) in each subgroup ($D^0$ - age < 70 and $D^1$ - age ≥ 70). The total number of samples as a function of uncertainty thresholds in are depicted with light colours in these plots.

## Appendix D. Definition and Calculation of Uncertainty

**Ensemble dropout (Smith and Gal, 2018):** An ensemble of N networks is (ex. 3 independent networks) trained using the same dataset split but different network weight initialization. During test time, the same input is passed through this ensemble with dropout at test time to collect M different samples (ex. 20 samples) for each network. This results in a total of T=M*N sample (ex. 60 samples) outputs across these networks.

**Entropy:** It is a measure of the informativeness of the model's predictive density function for each model output $\hat{y}_i$. It is defined as:

$$H[\hat{y}_i|x_i] = -\sum_{c=1}^{C} p(\hat{y}_i = c|x_i) \log\left(p(\hat{y}_i = c|x_i)\right)$$
$$\approx -\sum_{c=1}^{C} \left(\frac{1}{T}\sum_{t=1}^{T} p(\hat{y}_{i(t)} = c|x_i)\right) \log\left(\frac{1}{T}\sum_{t=1}^{T} p(\hat{y}_{i(t)} = c|x_i)\right). \tag{3}$$

where $C$ is the total number of class labels, and $p(\hat{y}_{i(t)} = c|x_i)$ denotes output softmax probability for class $c$ for sample $t$ (Gal and Ghahramani, 2016; Lakshminarayanan et al., 2017; Smith and Gal, 2018). High entropy implies a flatter probability distribution across classes, while low entropy implies a more peaky probability distribution. Lower entropy shows that model is more confident in its prediction of the output class. Predictive entropy measures both epistemic and aleatoric uncertainties (which will be high whenever either epistemic is high or aleatoric is high) (Gal, 2016; Gal et al., 2017). Here we only consider entropy for the classification and the segmentation task. The calculation of entropy for a regression task requires calculating a normalized histogram, a computationally intensive process.

**Sample Variance:** The simplest uncertainty measure, sample variance, is estimated by computing the variance across the $T$ samples collected Ensemble Dropout (Smith and Gal, 2018). For a regression task the variance in the output $\hat{y}_i$ for any input $x_i$, is defined as follows:

$$\text{Var}(\hat{y}_i) = \frac{1}{T}\sum_{t=1}^{T} \hat{y}_{i(t)}^2 - \left(\frac{1}{T}\sum_{t=1}^{T} \hat{y}_{i(t)}\right)^2. \tag{4}$$

where $\hat{y}_{i(t)}$ is a prediction for sample t. Sample variance can be more simply interpreted as a measure of model output consistency across different samples.

**Predicted Variance:** For Predicted Variance, during training, in addition to the labels, the weights of the network are also trained to produce the prediction variance $\hat{V}$ at the output. Please refer to (Kendall and Gal, 2017; Nair et al., 2020) for more details about predicted variance.

**Total Variance:** Sample variance measures epistemic (model) uncertainty, while predicted variance measures aleatoric (data) uncertainty. The summation of both sample variance and

predicted variance can give us the total variance (Gal, 2016; Kendall and Gal, 2017). We choose total variance for the regression task as it is computationally more feasible compared to the entropy for the regression task, and similar to the entropy, it also measures both aleatoric and epistemic uncertainties.

