# OpenReview forum: "Evaluating the Fairness of Deep Learning Uncertainty Estimates in Medical Image Analysis"
_MIDL.io/2023/Conference — MIDL 2023 Poster_

### Official Review · Reviewer_Czxr · 2023-02-02

**Confidence:** 3
**Preliminary Rating:** 4

**Summary:**

The work investigated the fairness of deep learning uncertainty estimation in different patient groups. The authors compared the fairness of three models/methods: a baseline model, a balanced model and a groupDro model, and they found that the two later methods achieved better fairness than the baseline. The validation was conducted on three different tasks.

**Strengths:**

1, The authors show an interesting topic to investigate the fairness of uncertainty estimation of deep learning models, which has not been studied before.

2, The experiments are comprehensive. The authors used three different representative tasks for validation, and compared the fairness of three different methods.


**Weaknesses:**

1, Some part of the method is not clear. I found it hard to understand the definition of the uncertainty estimation fairness. In my understanding, EM in eq. 1. is related to the model’s classification/segmentation performance, and EM in Eq. 2 should be some metrics for evaluating the uncertainty estimation quality, such as calibration and uncertainty-error overlap in the literature [1]. Why not using such metrics? [1] Jungo et al. Assessing reliability and challenges of uncertainty estimations for medical image segmentation, MICCAI 2019.

2, The authors defined FG based on the model’s prediction quality under different uncertainty thresholds. It’s not clear to me how tuning the uncertainty thresholds will affect the perdition performance. Does it mean only the prediction accuracy is calculated in the certain cases? This should be clarified and need more explanation.

3, FG is defined in Eq.2 but the results section did not show the comparison or distribution of FG scores.



**Deanonymize Review:**

no

**Detailed Comments:**

For the brain tumor segmentation task, the authors split the data set into two subgroups: large tumor and smaller tumors. This is not as good as split based on gender as used in the first task. It is usually more challenging to segment smaller tumors with high Dice scores. Therefore, most deep learning models may be biased to large tumors, which leads to unfairness caused by patient groups rather than the deep learning model. I think the authors should consider a different way for sub-group splitting.

**Paper Type:**

validation/application paper

**Questions To Address In The Rebuttal:**

1, The authors need to clarify the definition of the fairness and give the motivation.

2, It would be good to see how the FG scores distribute in the experimental results.

3, The rationality of splitting the BraTS dataset need to be clarified. Why not using other attributes such as gender or imaging centers?

---

### Official Review · Reviewer_tdRL · 2023-02-05

**Confidence:** 4
**Preliminary Rating:** 3

**Summary:**

The authors investigate the uncertainty estimation performance of bias-mitigation algorithms in the context of medical imaging. They defined the fairness gap as the group fairness metric and used the ensemble dropout model as the uncertainty quantification method. In the experiments, they performed skin lesion classification, brain tumor segmentation and AD clinical score regression. Here age and enhancing tumor volume are the sensitive attributes. They found that the uncertainty estimation ability is decreased with the increased bias reduction performance.

**Strengths:**

- The study is well motivated that uncertainty quantification is essential for medical tasks;
- Diverse datasets and tasks are introduced in this study;
- The paper is clear and overall well written;


**Weaknesses:**

- Fairness metrics are not compatible with each other. Perhaps the findings need to be validated by other metrics;
- Dropout model is a shortcut of Bayesian deep learning for uncertainty quantification. Other uncertainty estimation methods such as conformal prediction might be desired;
- We also can observe that there’s a fundamental tradeoff between fairness and precision. Perhaps this is a more important issue to tackle before talking about uncertainty estimation. Because it’s not appropriate to compare uncertainty when the two models are at different precision levels;
- The study only investigated age and tumor volume as the sensitive attributes. Also for each attribute, only two subgroups are divided without any explanation on why;


**Deanonymize Review:**

no

**Paper Type:**

validation/application paper

**Questions To Address In The Rebuttal:**

- Introduce more group fairness metrics beside fairness gap;
- Introduce uncertainty quantification method from different perspectives;
- Justify on why uncertainty estimation is an important issue to tackle when fair models are at a disadvantage precision compared to the vanilla one;
- Justify on why these two attributes are representative for the conclusion of the study and how the thresholds were chosen;

---

### Official Review · Reviewer_YXcH · 2023-02-06

**Confidence:** 3
**Preliminary Rating:** 2

**Summary:**

The paper investigates the fairness of deep learning (DL) models among different demographic subgroups in the field of medical image analysis. It’s been revealed in previous works that DL models can exhibit lower performance for diabetic retinopathy diagnosis, cardiac MR image segmentation and brain MR segmentation. While the fairness can be enhanced by re-balancing the dataset, the authors argue that estimating uncertainty in the predictions can further establish trust in clinical practice. The authors apply uncertainty quantification for population subgroups, and indicate that there is a lack of fairness among different subgroups, while the methods that enforce fairness achieves fairness at the cost of high uncertainty.

**Strengths:**

The paper investigates an interesting and critical problem for medical image analysis: the fairness of performances among demographic groups. Popular tasks, such as skin lesion classification, brain tumor segmentation and Alzheimer’s disease clinical score regression, have been used to evaluate the fairness, using both baseline, GRO (by reweighing the loss for each sub group) and balanced models (trained on balanced datasets). The paper is overall easy to follow and shows sufficient analysis results.


**Weaknesses:**

- The uncertainty u is not clearly defined, how is the uncertainty calculated and normalised?
In Fig. 1 and 3, what do the curves in lighter colours indicate?
- The relation between uncertainty and the fairness gap is hard to generalise, for example, in Fig. 1(c), GroupDRO-model does not always reduce the fairness gap with few predictions filtered, such as in the third column, and in Fig. 3, the relation between uncertainty and fairness gap is different as in column 2 and columns 1&3.
- Some minor issues, in page 5, “Figure-6(c) shows that the GroupDRO-Model gives better …”, should be Figure 1(c).
- The paper is an empirical analysis focused on evaluating the fairness gap for different sub-groups and indicates the relation between fairness gap and uncertainty. The strength is somewhat limited as only one attribute, age, is evaluated, while other attributes, such as gender and race.


**Deanonymize Review:**

no

**Paper Type:**

validation/application paper

**Questions To Address In The Rebuttal:**

The paper explores the issue of fairness of deep learning models for medical images, which is a critical problem if the models are to be applied in clinical practice. The paper is organised as an empirical analysis paper, and investigates the fairness gap and uncertainty on several well-known tasks in medical image analysis.

To strengthen the paper, a few concerns needs to be addressed:

- the definition and calculation of uncertainty $u$ is not clear;
- the relation between fairness gap and uncertainty seems hard to generalise for different classes in each task.
- please provide some insights on the fairness in terms of other important attributes such as race and gender.

---

### Meta-Review · Area_Chair_wGFE · 2023-02-26

**Recommendation:** Accept (Poster)
**Confidence:** 3

**Metareview:**

This paper remains borderline, after very thorough rebuttal, including many additions to the paper (for example, including experiments on more attributes). The reviews were thoughtful, and the answers from the authors were extensive and clear.

I appreciate the author's clarification that the paper is not focused on showing that fairness mitigation work largely leads to losses in the ability to understand uncertainty, which has a bearing on clinical distribution. As far as I can tell, the paper aims to show this empirically. While not the standard paper in our community, I do appreciate these sort of insights and believe they can be quite valueable.

However, there are some aspects that are worrysome, I believe. One aspect that remains unclear to me, and particularly important in this sort of paper, is quite how general this conclusion is -- as raised by some reviewers, the limited attributes, methods (mostly deep dropout), and hyperparameter choices (e.g thresholding). I appreciate that the authors evaluated more attributes for the rebuttal (although disagree that the original page length limitations is sufficient for a reason to not have ran these from the start -- while limited, the page count is quite substantial and appendices are allowed). Nevertheless, the conclusion is still limited to dropout and other certain choices, and thus the generality of the insight is questionable. While dropout is often used, it's by no means the only general strategy in uncertainty.

Overall, I think there is just enough insight here to allow for interesting discussion with the community, and am leaning slightly towards acceptance. The authors need to emphasize in the camera ready that their conclusion is limited to one (perhaps prevalent) set of choices, and to highlight in the main text the totality of attributes analyzed (even if these are only included in the appendix, post-rebuttal). Please also make sure to include all the clarifications required by reviewers (e.g. reasoning for splitting the BraTS dataset)